

# The first leech body fossil predates estimated hirudinidan origins by 200 million years

Danielle de Carle[1], Rafael Eiji Iwama[2], Andrew J. Wendruff[3], Loren E. Babcock[3] and Karma Nanglu[1,4,5]

[1] Department of Ecology & Evolutionary Biology, University of Toronto, Toronto, Ontario, Canada
[2] Departamento de Genética e Biologia Evolutiva, Instituto de Biociências, Universidade de São Paulo, São Paulo, São Paulo, Brasil
[3] Orton Geological Museum, School of Earth Sciences, The Ohio State University, Columbus, Ohio, United States of America
[4] Museum of Comparative Zoology & Department of Organismic and Evolutionary Biology, Harvard University, Cambridge, Massachusetts, United States of America
[5] Department of Earth and Planetary Sciences, University of California, Riverside, Riverside, California, United States of America

## ABSTRACT

Clitellata is a major annelid clade comprising oligochaetes (*e.g.*, earthworms) and hirudineans (*e.g.*, leeches). Due to their scant fossil record, the origins of clitellates, particularly Hirudinea, are poorly known. Here, we describe the first leech body fossil, *Macromyzon siluricus*, gen. et sp. nov., from the Brandon Bridge Formation (Waukesha Lagerstätte). This fossil, which is preserved in exceptional detail, possesses several hirudinean soft-tissue synapomorphies–including a large sucker at the posterior end and sub-divided segments–and phylogenetic analyses resolve *Macromyzon siluricus* as a stem leech. Its age, 437.5–436.5 Ma, is consistent with early age estimates for the origin of clitellates, and predates molecular-clock-based estimates of hirudinidan origins by at least 200 million years. These findings suggest that the earliest true leeches were marine and that, contrary to prevailing hypotheses, were unlikely to have fed on vertebrate blood.

## INTRODUCTION

Annelida, a speciose and morphologically disparate phylum within the animal group Lophotrochozoa, possesses a fossil record stretching back to the Cambrian. This consists primarily of polychaetes, non-biomineralized examples of which are recorded from deposits of exceptional fossil preservation such as the Burgess Shale in British Columbia, Canada (*Conway Morris, 1979*; *Nanglu & Caron, 2018*); Sirius Passet (Buen Formation) in North Greenland (*Conway Morris & Peel, 2008*); and both the Chengjiang (Maotianshan Shale) and Guanshan Formation biotas of South China (*Liu et al., 2015*; *Han et al., 2019*). The fossil record of scolecodonts (polychaete jaw elements) indicates that the first major polychaete radiation occurred during the Ordovician (*Hints & Eriksson, 2007*).

Corresponding author
Danielle de Carle,
danielle.decarle@utoronto.ca

These fossils, in conjunction with novel modes of analysis and more robust molecular datasets (*Parry et al., 2016*; *Weigert & Bleidorn, 2016*; *Nanglu & Caron, 2018*), are helping to clarify some of the broad phylogenetic interrelationships among annelids; however, the origins of the major annelid clade Clitellata (earthworms, leeches, and their relatives) remain uncertain. In part, this is due to the uncertain phylogenetic placement of Clitellata (*Weigert & Bleidorn, 2016*), but, a more significant impediment is the exceedingly poor published fossil record of clitellates (*Parry, Tanner & Vinther, 2014*; *Bomfleur et al., 2015*). This paucity is unsurprising given that clitellate anatomy consists almost entirely of non-biomineralized tissue. Taphonomic studies have shown that polychaetes decay rapidly post-mortem: only their sclerotized jaws possess high decay resistance (*Briggs & Kear, 1993*). Although branchiobdellidans and some leeches possess hardened jaws, these structures are absent in the vast majority of clitellates, which reduces the likelihood that they will fossilize under most aqueous depositional circumstances.

Clitellata comprises roughly one-third of all extant annelid species, and they can be found in nearly every conceivable habitat on Earth (*Sawyer, 1986*; *Martin et al., 2008*; *Sket & Trontelj, 2008*). The group is characterized by the possession of a clitellum, a glandular region in the anterior part of the body which secretes a cocoon into which eggs are deposited (*Sawyer, 1986*). They are readily distinguished from other annelids by their lack of parapodia, the reduction or lack of chaetae, and the presence of certain autapomorphic internal reproductive characteristics (*Brinkhurst, 1982*; *Rouse & Fauchald, 1995*; *Westheide, 1997*). Clitellates are generally split into two groups: the paraphyletic Oligochaeta (earthworms and their relatives), and Hirudinea (Branchiobdellida, the crayfish worms; Acanthobdellida, the hook-faced fish worms; and Hirudinida, the true leeches) (*Tessler et al., 2018*; *Erséus et al., 2020*).

Today, clitellates are of great ecological, economic, and evolutionary importance. Non-hirudinean clitellates are among the most important ecosystem engineers in aquatic and terrestrial sedimentary environments (*Lavelle et al., 1997*), and hirudinean taxa fill a broad diversity of ecological niches as parasites, commensal symbionts, predators, and disease vectors (*Sawyer, 1986*; *Martin et al., 2008*). With the acquisition of a caudal sucker, and reduction of the clitellum, chaetae, and internal segmentation, hirudineans represent a unique annelid body plan (*Purschke et al., 1993*). Furthermore, acanthobdellidans and leeches represent a novel radiation of parasitic annelids, and as a result, the latter have been used as medical tools for thousands of years, continuing to the present day. Based primarily on morphological evidence and ancestral state estimation, most authors have converged on the hypothesis that the earliest clitellates were freshwater organisms, but the precise timing of the transition to a terrestrial habitat is not known (*Manum, Bose & Sawyer, 1991*; *Rousset et al., 2008*; *Erséus et al., 2020*).

*Shcherbakov et al. (2020)* provided a comprehensive summary of the clitellate fossil record, which largely consists of poorly preserved oligochaetes; these authors do not recognize any leech body fossils. A specimen putatively labelled (but not formally described) as "?Leech" by *Mikulic, Briggs & Kluessendorf (1985a)* and cited by *Briggs (1991)* lacks necessary characteristics that are diagnostic of leeches (*de Carle, 2022*), and has since been referred to the Cycloneuralia (*Braddy, Gass & Tessler, 2023*). Exceptionally preserved

leech cocoons have been documented in deposits dating to the Triassic (*Manum, Bose & Sawyer, 1991*; *Bomfleur et al., 2012*; *Steinthorsdottir, Tosolini & McElwain, 2015*). These fossil cocoons, which have been categorized into four distinct genera, exhibit a range of morphologies, all of which are consistent with Hirudinea (*Manum, Bose & Sawyer, 1991*; *McLoughlin et al., 2016*). Beyond this, it is difficult to associate any particular cocoon morphology with a corresponding hirudinean taxon. Cocoon building is ubiquitous within Hirudinea–and Clitellata more broadly–indicating that this behaviour predates, or at least coincides with, the origins of the clade (*Manum, Bose & Sawyer, 1991*; *Bomfleur et al., 2012*). Although cocoon morphology is consistent within some higher-order hirudinean taxa (*e.g.*, Hirudinidae, Erpobdellidae, Glossiphoniidae), little information is available regarding the cocoons produced by other groups (*e.g.*, Branchiobdellida). Spermatozoa consistent with those of extant Branchiobdellida have been preserved within some of these cocoons dating to the Eocene of Antarctica (*Bomfleur et al., 2015*; *McLoughlin et al., 2016*). Ultimately, however, owing to the variability in the cocoons and spermatozoa of extant hirudineans (*Bomfleur et al., 2015*; *McLoughlin et al., 2016*), we are left with little definitive insight into the evolution, ecology, and anatomy of the animals that produced these trace fossils.

The scant fossil record of Clitellata leaves a number of questions unresolved. Although molecular clocks have been used to estimate divergence times for Clitellata, Hirudinea, and true leeches (*e.g.*, *Edgecombe et al., 2011*; *Erwin et al., 2011*; *Erséus et al., 2020*), a dense sampling of fossil calibration points is needed to infer divergence times precisely and accurately (*Warnock, Yang & Donoghue, 2017*). A better understanding of early clitellate history would elucidate the timing of important evolutionary events including terrestrialization, development of the hirudinean body plan, and the origins of parasitism.

Here, we report the first fossil leech from the lower Brandon Bridge Formation (Silurian: Llandovery, Telychian) of Wisconsin, USA. The Brandon Bridge biota is diverse, including the remains of biomineralizing, non-biomineralizing and lightly biomineralizing organisms (*Mikulic, Briggs & Kluessendorf, 1985a*, *Mikulic, Briggs & Kluessendorf, 1985b*; *Wendruff et al., 2020a*, *2020b*). This array of anatomies with varying taphonomic potentials includes even the most labile tissues found in leeches and other non-sclerotized animals (*Briggs, 1991*; *Briggs & Kear, 1993*; *Saleh et al., 2020*). The exceptionally preserved form described herein shows evidence of synapomorphic hirudinean characters–including a large caudal sucker and segments sub-divided into annuli–and does not bear hallmarks of other vermiform taxa, such as palaeoscolecids. This new species allows for a re-evaluation of the timing of clitellate origins, as well as new consideration of ancestral hirudinean ecology.

## MATERIALS AND METHODS

### Specimen preparation and imaging

Our fossil specimen was exposed using a freeze-thaw technique to split the dolostone along the bedding-plane-parallel fossil. The tissue has been preserved as a thin organic film, and lacks any secondary phosphate or other diagenetic overprinting; compare with other fossils

in *Wendruff et al. (2020a*, *2020b)*. It is deposited in the collections of the University of Wisconsin-Madison Geology Museum, accession number UWGM 7056.

The specimen was photographed with a Canon EOS Rebel T3i Digital SLR using a Canon MP-E 65 mm macro lens and full spectrum lighting. Images were stitched using Adobe Photoshop CC 2015. Measurements were made using ImageJ ver. 1.53, and figures were assembled using Adobe Illustrator ver. 19.2.0.

## Nomenclatural acts

The electronic version of this article in Portable Document Format (PDF) will represent a published work according to the International Commission on Zoological Nomenclature (ICZN), and hence the new names contained in the electronic version are effectively published under that Code from the electronic edition alone. This published work and the nomenclatural acts it contains have been registered in ZooBank, the online registration system for the ICZN. The ZooBank LSIDs (Life Science Identifiers) can be resolved and the associated information viewed through any standard web browser by appending the LSID to the prefix http://zoobank.org/. The LSID for this publication is: urn:lsid:zoobank.org: pub:D5B80E8C-F0F8-4FDC-A617-29E31F6745AE. The online version of this work is archived and available from the following digital repositories: PeerJ, PubMed Central SCIE and CLOCKSS.

## Phylogenetic analyses

As a complement to our morphological investigation, we performed phylogenetic analyses using Bayesian inference and maximum parsimony optimality criteria. Our phylogenetic matrix (File S1) was adapted from *Ou et al. (2017)*, which was designed to accommodate a wide sampling of metazoan taxa. A broad selection of body plans was deemed necessary to accommodate the potential variety of affinities of this new taxon, as further detailed in the "Remarks" section below. Non-bilaterian taxa were removed, as *Macromyzon siluricus* is clearly bilaterian in nature. Four further groups–Xenacoelomorpha, Gastrotricha, Gnathifera, and Chaetognatha–were removed because their systematic positions remain debated based on both morphological and molecular data (*Caron & Cheung, 2019*; *Simakov et al., 2013*; *Philippe et al., 2019*; *Kapli & Telford, 2020*; *Cannon et al., 2016*). Their inclusion could therefore skew resulting topologies without providing meaningful data about the phylogenetic position of *Macromyzon*.

A number of taxa were added to the character matrix to test our hypotheses of *Macromyzon*'s affinities. The polychaete *Capitella* was added, as the family Capitellidae has been hypothesised to be among the closest relatives of Clitellata (*Weigert & Bleidorn, 2016*). *Eunice* and *Harmothoe* were added to accommodate for the possibility that *Macromyzon siluricus* may have a phylogenetic affinity with a polychaete body plan (character codings based on *Rouse, Pleijel & Tilic, 2022*). The fossil forms *Esconites* and *Dryptoscolex* were added as their relationships to the polychaete families Eunicidae and Polynoidea are relatively well established (character codings based on *Parry et al., 2016*). Based on external similarities, five clitellate taxa were added: the subclass Oligochaeta, and the three hirudinean orders Acanthobdellida, Branchiobdellida and Hirudinida (the true

leeches, represented by *Hirudo* and *Glossiphonia*) (character codings based on *Holt (1965)*, *Sawyer (1986)*, *Purschke et al. (1993)*, *Weigl (1994)*, *Bielecki et al. (2014)*, *de Carle et al. (2022)*). We have added the fossil palaeoscolecid *Cricocosmia*, to corroborate our hypothesis that *Macromyzon siluricus* lacks any palaeoscolecid affinity (character codings based on *Cong et al. (2017)*).

Addition of new taxa required that we add new characters to the phylogenetic matrix. In total, we added 19 new characters (char. 56–74; File S1): the majority of which are designed to infer annelid relationships (*e.g.*, segmentation patterns, and morphology of jaws, clitellum, and reproductive anatomy). Notes on these characters and their coding are included in the Supplementary Information. Finally, invariant characters were deleted from the character matrix to avoid overestimation of branch lengths that results from acquisition bias (*Lewis, 2001*).

Phylogenetic analyses were performed under Bayesian inference in MrBayes ver. 3.2.7a (*Ronquist & Huelsenbeck, 2003*) and maximum parsimony in TNT ver. 1. 5 (*Goloboff, Farris & Nixon, 2008*). Bayesian analysis used the mkv+gamma model, and a symmetrical Dirichlet hyperprior for all characters. Tracer ver. 1.7.1 (*Rambaut et al., 2018*) was used to determine whether the simulations had converged (*i.e.*, effective sample sizes for all parameters >200; sampling reached stationarity; average standard deviation of split frequencies <0.01). Parsimony heuristic searches ran for 10,000 iterations each with five rounds of ratcheting. Support values were calculated from 1,000 rounds of bootstrapping.

The resulting Bayesian and parsimony trees are shown in Figs. S1 and S2. Raw output from these analyses (Files S2, S3), as well as the complete phylogenetic matrix and commands used to run MrBayes (File S1), are included as supplementary information.

Because there is some uncertainty surrounding the interpretation of annulation and segmentation in *Macromyzon*, we ran two sets of supplementary analyses with variable coding for characters 61 (segments externally subdivided into annuli) and 62 (annulation pattern in midbody segments). In our initial analyses, *Macromyzon* is coded as "present" and "triannulate" for these characters, respectively. In the first supplemental analysis, both characters were coded as unknown for *M. siluricus* and in the second, the characters were coded as "present" and "unknown", respectively. The results of these analyses are shown in Figs. S3–S6 and raw output is included as Files S4–S7.

## RESULTS

### Systematic palaeontology
Phylum: Annelida, Lamarck 1809
Class: Clitellata, Michaelson 1919
Subclass: Hirudinea, Lamarck 1818
Genus: *Macromyzon* gen. nov.
urn:lsid:zoobank.org:act:56E87D9F-D838-4FFA-AC47-57825B46EAED
Type Species: *Macromyzon siluricus* sp. nov.
urn:lsid:zoobank.org:act:8F3D2E52-2AE0-4F6C-9C8B-0AE1C62EA423

## Etymology

*Macromyzon* (gender: masculine), from Greek, *makros*, "large" + *myzon*, "sucker" in reference to the large caudal appendage. The specific epithet refers to the Silurian age of the fossil.

## Material examined

Holotype: University of Wisconsin Geology Museum, specimen number: UWGM 7056.

Locality: Waukesha Lime and Stone Company, west quarry, north of State Highway 164, Waukesha, Wisconsin, USA.

Stratigraphic occurrence: Lower part of the Brandon Bridge Formation (Silurian: Llandovery, Telychian; *Mikulic & Kluessendorf, 1999*).

Stratigraphic context and taphonomy: The fossil material examined here was collected from the Waukesha Lime and Stone Company west quarry in Waukesha, Wisconsin, USA (*Mikulic, Briggs & Kluessendorf, 1985a*, *1985b*; *Wendruff et al., 2020a*). It is from the lower part of the Brandon Bridge Formation (Silurian: Llandovery, Telychian), better known as the Waukesha Lagerstätte. Taxa from this locality were first reported by *Mikulic, Briggs & Kluessendorf (1985a*, *1985b)*. Subsequently, various taxa have been discussed in a number of articles, and summarized by *Wendruff et al. (2020a)*. These include a diverse array of arthropods, including a new scorpion (*Wendruff et al., 2020b*, although see *Anderson et al. (2021)* for an alternative interpretation), a vermiform arthropod *Archeronauta* (*Pulsipher et al., 2022*), three species of the phyllocarid *Ceratiocaris* (*Jones, Feldmann & Schweitzer, 2015*), and trilobites (*Wendruff et al., 2020a*; *Randolphe & Gass, 2024*).

The taxonomic composition of the biota and some taphonomic pathways leading to exceptional fossil preservation were reviewed by *Wendruff et al. (2020a)*, who also reviewed available biostratigraphic information bearing on the age of the Waukesha Biota. Conodonts provide the most precise biostratigraphic guides known from the lower Brandon Bridge Formation near Waukesha, Wisconsin (*Mikulic & Kluessendorf, 1999*). According to *Kleffner et al. (2018)*, the euconodont *Pterospathodus eopennatus*, the eponymous indicator of the *P. eopennatus* Superzone, is present. This guide fossil constrains the deposit to the Telychian Stage (Silurian System, Llandovery Series, approximately 437.5–436.5 Ma).

Palaeoenvironmentally, the Brandon Bridge deposit represents a tropical, carbonate marine platform setting (*Kluessendorf, 1990*; *Wendruff et al., 2020a*). Remains of organisms were transported into intertidal to supratidal sedimentary traps associated with karstic topography adjacent to the shoreline. Some remains were quickly covered by cyanobacterial mats. Early diagenetic activity within the mats helped preserve some bodily remains. Replication by thin calcium phosphate (apatite) encrustation is common, but some non-biomineralised parts are preserved as carbon films.

## Diagnosis

Hirudinean having a vermiform-sublanceolate shape, truncated anteriorly, width expanding toward posterior, reaching maximum width slightly anterior of margin. Posterior margin approximately twice the width of anterior margin. Body segmented, with

regularly spaced, sexannulate divisions. Posterior terminates in a large caudal sucker. The genus is monospecific; the diagnosis applies to both genus and species.

## Description

The holotype, and only known specimen, of *Macromyzon siluricus* (Figs. 1A–1C) is 51 mm in length. As preserved, the specimen is twisted near midlength, approximately 19 mm from the anterior end; the narrower anterior portion exposes the dorsal side, and the wider posterior portion exposes the ventral side.

Outline is vermiform-sublanceolate, truncated at anterior end; width expands toward posterior, reaching a maximum width of 18.9 mm at the posterior end somewhat forward of the posterior terminus, then narrowing slightly toward posterior margin. Integument divided into numerous, narrow, sexannulate divisions, subequal in length (sag.); divisions covered with semicircular protrusions, which we interpret as metameric circular organs (Figs. 1C–1E). A large ellipsoid structure, which we interpret as a caudal sucker, encompasses approximately the entire width of the posterior end (Figs. 1A–1B, 2A). It is preserved more darkly in comparison with the surrounding material, and thin, concentric striations around the caudal sucker give it a wrinkled appearance.

## Remarks

External evidence of segmentation, coupled with the lack of a chitinous cuticle, identifies *Macromyzon siluricus* gen. et sp. nov. as an annelid. The lack of a chitinous cuticle is inferred from comparison with chitin-bearing arthropods in the Waukesha Lagerstätte (*Wendruff et al., 2020a*). As preserved, chitinous cuticle has a thickness up to about 0.5 mm, enough to result in a distinct elevation difference between the fossil and the adjacent matrix. Commonly, chitin also is encrusted with secondary calcium phosphate. Neither of these conditions is shown in *M. siluricus*. Although segmentation is also a feature of arthropods, *M. siluricus* is divided into thinner, non-sclerotized segments compared to the segmentation in co-occurring arthropod taxa. Furthermore, there is no evidence of jointed or lobopodous limbs associated with these segments that would suggest panarthropod affinity. Finally, the segments of *Macromyzon* do not appear to have any features consistent with limbs that may have been preferentially lost due to taphonomic factors. The combination of a prominent caudal sucker, lack of external appendages, and the subdivision of segments into annuli supports an affinity with the Clitellata, specifically the hirudineans (*Sawyer, 1986*; *Purschke et al., 1993*; *Westheide, 1997*). Some polychaete lineages have lost parapodia and chaetae independently of the clitellates, and subdivided segments are present in many members of Scalibregmatidae and Opheliidae, but no non-hirudinean annelids possess suckers (*Rouse, Pleijel & Tilic, 2022*).

The large caudal sucker of *M. siluricus*, which occupies approximately 3/4 of the posterior body width (Figs. 1A–1B, 2A), is similar to the well-developed caudal suckers that characterize members of Hirudinea (Figs. 1F–1G, 2) (*Sawyer, 1986*). Depending on their location on the body, caudal suckers can be considered either terminally (*e.g.*, Fig. 2F) or ventrally positioned (*e.g.*, Fig. 2G). Terminally positioned caudal suckers may be directed anteriorly or folded posteriorly on the end of the posterior-most somite, whereas

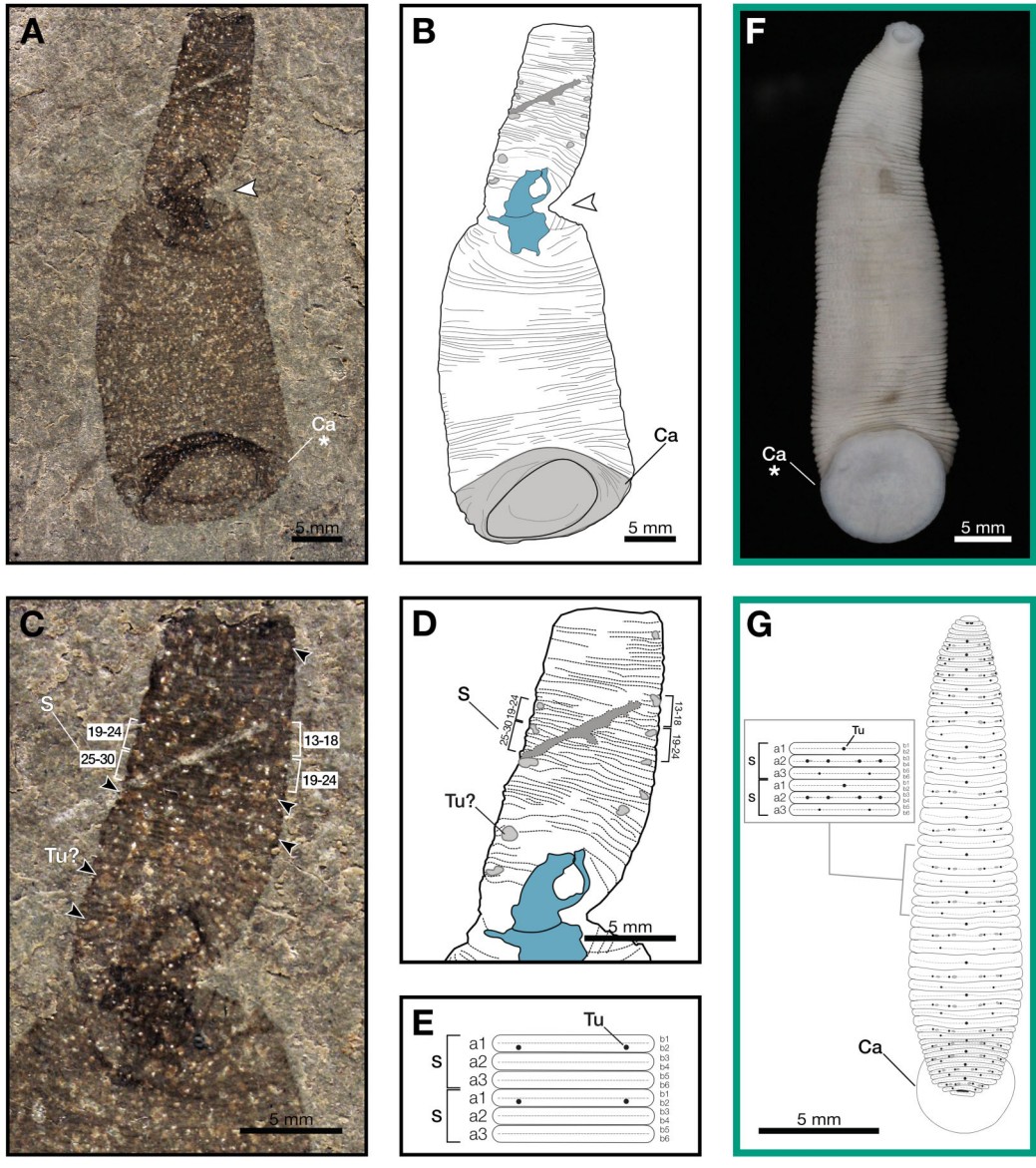

**Figure 1** *Macromyzon siluricus* **gen. et sp. nov. from the Lower Brandon Bridge Formation (Waukesha Lagerstätte), Silurian (Llandovery: Telychian), Waukesha, Wisconsin, USA.** (A) Holo-type specimen UWGM 7056. (B) Schematic of the external morphology of *Macromyzon siluricus* based on the holotype. (C) Detail of the anterior region, dorsal view, showing sexannulate segments with annuli numbered; black arrows indicate putative tubercles. (D) Schematic of the anterior region showing tubercles in light grey and sexannulate segments with annuli numbered. (E) Schematic of segmentation pattern for *M. siluricus*. Green borders indicate extant Hirudinida introduced for comparison: (F) Ventral view of *Myxobdella sinanensis* (Zoological Collection of Kyoto University, specimen KUZ Z1794); photo by T. Nakano. (G) Schematic of *Haementeria lutzi* (dorsal view) with inset showing the species' segmentation pattern. This specimen is deposited in the collections of the Museum of Zoology of the University of São Paulo (MZUSP 0026). Abbreviations: Ca, caudal sucker; S, segment; Tu, tubercles; Tu?, putative tubercles (metameric circular organs). White arrow indicates mid-body torsion, the point of torsion is shown in blue on schematics; breakage in the specimen is indicated in dark grey. Detail of starred features is shown in Figs. 2A, 2B.

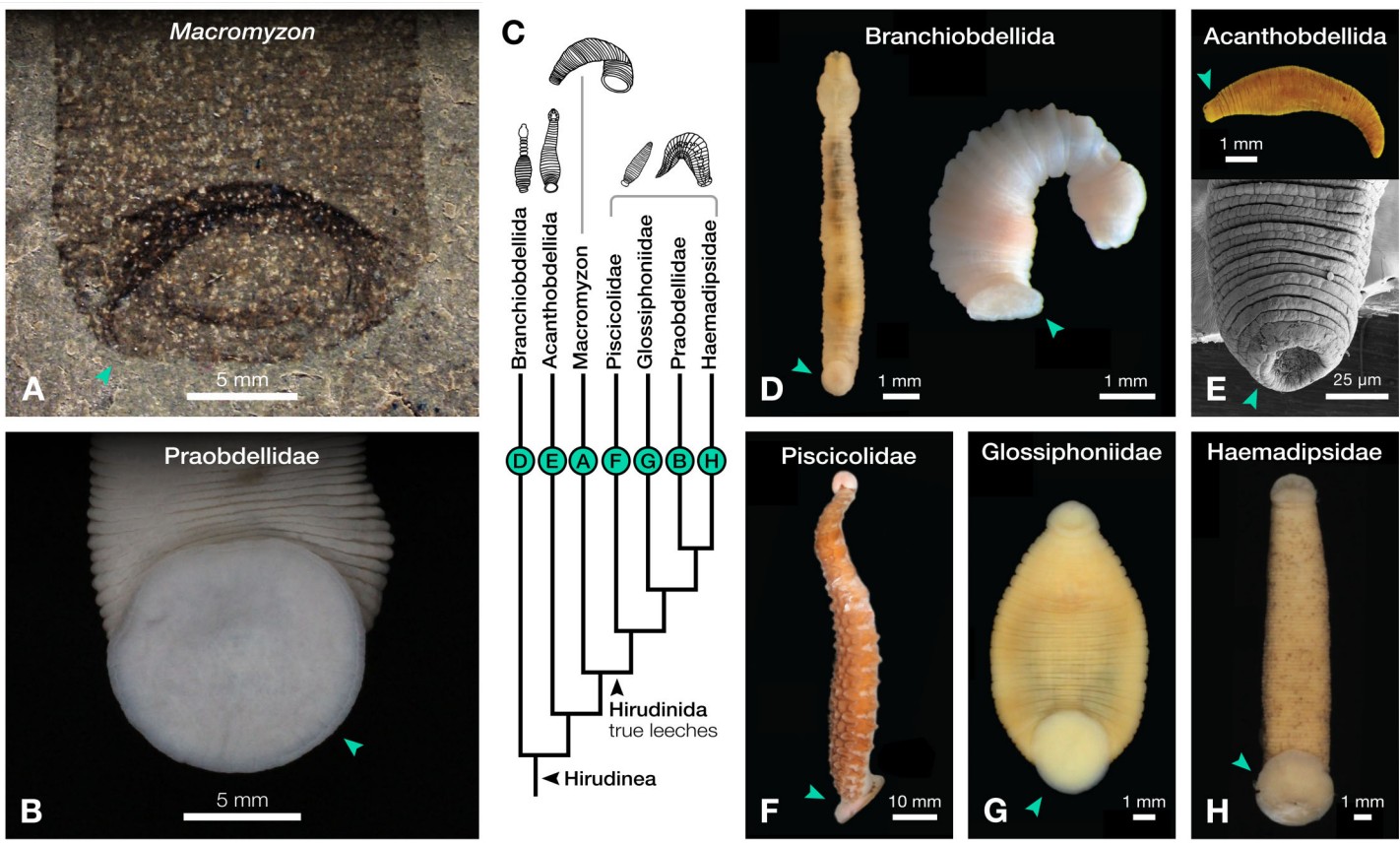

**Figure 2** Comparison of caudal suckers (green arrows) in *Macromyzon* and extant Hirudinea. (A) Detail of the caudal sucker in *Macromyzon siluricus*, whole specimen shown in Fig. 1A. (B) Detail of the caudal sucker in the hirudiniform leech *Myxobdella sinanensis* (Zoological Collection of Kyoto University, specimen KUZ Z1794), whole specimen shown in Fig. 1F. (C) Cladogram showing relationships between taxa represented in this figure. (D) Branchiobdellidan species *Cambarincola* aff. *okadai* (left; National Museum Cardiff, specimen NMW.Z.2014.004) and *Triannulata magna* (right; Museum National d'Histoires Naturelles, specimen MNHN-HEL656); photos by *James et al. (2015)* and *Parpet & Gelder (2020)*. (E) Full body (top) and scanning electron micrograph (bottom) of the acanthobdellidan *Acanthobdella peledina*; images by P. Swiątek. (F) The oceanobdelliform leech *Pterobdellina vernadskyi*; photo by *Utevsky, Solod & Utevsky (2021)*. (G) The glossiphoniiform leech *Torix* sp. (Zoological Collection of Kyoto University, specimen KUZ Z4325); photo by T. Nakano. (H) The hirudiniform leech *Haemadipsa japonica* (Zoological Collection of Kyoto University, specimen KUZ Z4324); photo by T. Nakano.

ventrally positioned suckers are attached to the ventral surface of the body. The caudal sucker of *M. siluricus* is in a terminal position, and the width of the posterior region at the point of attachment of the caudal sucker to the body is uniform. The sucker is folded toward the venter, obscuring some of the underlying annuli on the posterior-most segments, and resulting in the constricted appearance of the partially obscured segments (Figs. 2A–2B). Many modern leech lineages possess such characteristics–*e.g.*, Praobdellidae, Haemadipsidae, and some members of Piscicolidae (*Sawyer, 1986*; *Utevsky, Solod & Utevsky, 2021*)–as do branchiobdellidans and acanthobdellidans (Figs. 2D–2E) (*James et al., 2015*; *Parpet & Gelder, 2020*; *de Carle et al., 2022*). Terminal caudal suckers and a sublanceolate body shape are common features of leeches belonging to the family Glossiphoniidae (Fig. 2G) (*Sawyer, 1986*).

In Hirudinea, segments are sub-divided into rings, or annuli, which do not correspond to internal segmentation. Externally, furrows between each annulus are shallower than those demarcating somites. Each somite is divided into the same number of annuli along the length of the body, and annuli comprising a somite often exhibit metameric patterns of ornamentation, pigmentation, or papillation (*e.g.*, Fig. 1G). The holotype of *M. siluricus* shows external signs of segmentation as well as annuli. On the basis of regular patterns of variation in the apparent depth of the furrows between the annuli, each annulus that is externally visible (Figs. 1C–1E) seems to be secondarily sub-divided on both the dorsal side (expressed anteriorly) and ventral side (expressed posteriorly); this is particularly evident between annuli 13 and 26 (Fig. 1C, 1D).

Annulation patterns vary significantly between the three orders of Hirudinea. In extant Hirudinida, segments are divided into three primary annuli, which themselves may be subsequently divided into secondary, tertiary, or quaternary annuli (*Sawyer, 1986*). Typically, each annulus within a segment is uniform in height (Figs. 1D, 1E, 1G). Acanthobdellidan segments are invariably quadrannulate with only the third annulus sub-divided, resulting in some annuli being taller than others within the same segment (Fig. 2E). The majority of branchiobdellidans have biannulate segments, with the anterior annulus being markedly taller than its posterior counterpart (Fig. 2D). Unlike members of Acanthobdellida and Hirudinida, the annuli of Branchiobdellida are not secondarily subdivided (*Sawyer, 1986*), and typically flare out anteriorly creating a distinctly trapezoidal profile. The number and uniform height of annuli visible in *Macromyzon* suggests that the segments are most similar to those of Hirudinida. Specifically, it appears that segments are primarily triannulate, with each primary annulus being subdivided, resulting in a total of six annuli per segment.

This leech-like annulation pattern is further supported by the presence of metameric circular organs on the anterior end, on every sixth annulus. Eleven of these metameric circular organs can be distinguished by average size, position, and metameric distribution. They are distributed on the left and right margins of the anterior (dorsal) region at annuli 3, 13 and 14, 19 and 20, 27 and 28, 33 and 34, 37 and 38, 46 and 47, and 51 (Figs. 1C–1E). Six of them are paired. External markers of the segmentation pattern are common in leeches (Fig. 1G). The most common ornaments found on leech integument are tubercles and sensillae, both of which occur in metameric patterns that repeat on each midbody segment. If the metameric circular features present in *M. siluricus* are not taphonomic artefacts, it is possible that they represent such organs.

As with many fossilized, soft-bodied, vermiform taxa (*Nanglu & Caron, 2018*), it may be difficult to definitively distinguish the dorsal and ventral sides of an animal, unless there are features strongly associated with either side (such as eyes; *Yang et al., 2024*). Fortunately, in the case of *Macromyzon*, we have two such features. First, the caudal suckers of extant leeches may be oriented ventrally or posteriorly, but never dorsally. The caudal sucker of *M. siluricus* appears to be posteriorly oriented, but folded ventrally, as discussed above (Figs. 2A, 2B, 2D–2H). Second, in extant, dorsoventrally flattened leeches, tubercles and sensillae are only found on the animal's dorsum (*Sawyer, 1986*). At the twist point, there is a pinching of the outline of the specimen and a deviation of the annuli from

uniformly parallel to concentric rings. The fact that these putative tubercles are only observed anterior to the midbody constriction (Figs. 1B–1D), combined with the orientation of the caudal sucker, and the directionality of the annuli in this region of the fossil, support the hypothesis that this constriction represents a full twist in the specimen. The right lateral edge of the specimen appears to cross over the body, with the left lateral edge crossing behind, such that the anterior portion of the fossil displays the dorsum, and the posterior portion of the fossil displays the specimen's venter (Figs. 1A, 1B). There is also a dark, irregularly shaped area where the cuticle is thickened and compacted. Furrows between the annuli are more pronounced on the dorsal side (anterior to the twist) than they are on the ventral side (posterior to the twist). These differences are consistent with dorso-ventral differences in annulation and ornamentation patterns present in many extant leeches.

In leeches, the number of annuli is highly variable among species, and sexannulate segments, like those observed in *Macromyzon*, are common (*Sawyer, 1986*). In total, the new species has 109 visible annuli amounting to 15 visible segments. This number does not include annuli that are wholly or partially obscured by the torsion, nor does it include the fused segments that comprise the sucker, which typically consists of five fused segments in extant leeches. As is in most Hirudinea, no structure resembling a clitellum is visible. In extant hirudineans, the clitellum is located anteriorly, in segments 10–12 (*Sawyer, 1986*); therefore, presuming that *Macromyzon* has a fixed number of segments, and that the number is similar to other hirudineans ($n = 15–33$) (*Purschke et al., 1993*), the number of visible segments ($n = 15$) in our fossil is sufficient to claim that an externally visible clitellum is absent in *M. siluricus*.

In addition to the diagnostic hirudinean characteristics noted above (*i.e.*, caudal sucker, subdivided annuli, and absence of any lateral appendages), *M. siluricus* notably lacks defining features of any other plausible vermiform clade. The apparent absence of chaetae indicates that *Macromyzon* is not a polychaete, as polychaetes with chaetae preserved are present in the Waukesha Biota (*Wendruff et al., 2020a*). Because they are reduced in both size and number, chaetae of oligochaetes might be less likely to fossilize; however, the large and well-preserved caudal sucker of *M. siluricus* argues against an oligochaete affinity. Some parasitic flatworms (platyhelminthes) also have large caudal host-attachment structures called opisthaptors, which may resemble a sucker. However, the lack of hooks and other attachment substructures characteristic of opisthaptors, in addition to the presence of distinctive annuli on *M. siluricus*, argue against a flatworm affinity.

Although the surface texture of *M. siluricus* bears some superficial similarity to palaeoscolecids, morphological and taphonomic evidence dismiss any affinity between these taxa. Palaeoscolecids have elongate bodies which characteristically curl ventrally inwards post-mortem (*Topper et al., 2010*; *García-Bellido, Paterson & Edgecombe, 2013*) (Figs. 3A–3C), unlike the sublanceolate and straight pose of *Macromyzon* (Figs. 1A, 1B). Palaeoscolecids also have thickened cuticles with an elaborated series of hardened sclerites comprising a comprehensive scleritome; imagery of the palaeoscolecid *Hadimopanella* shows these sclerites in greater detail (Fig. 3D). These features are clearly visible in other, currently undescribed, palaeoscolecids from Waukesha (*Wendruff et al., 2020a*)

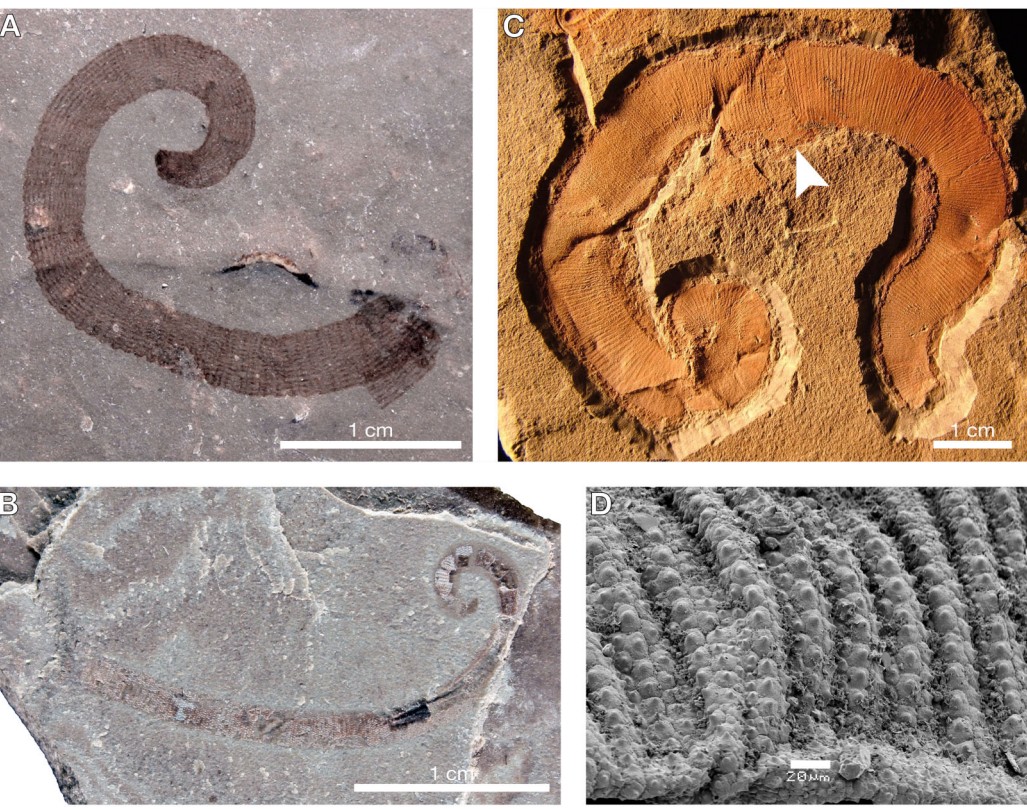

**Figure 3 A selection of palaeoscolecids for comparative purposes, refuting any affinity between** *Macromyzon* **and vermiform ecdysozoans.** (A) An undescribed palaeoscolecid from Waukesha showing a toughened, plate-like arrangement of the cuticle and scleritome. (B) Another undescribed palaeoscolecid from Waukesha, with tightly packed, regularly arranged, dome-shaped sclerites. A and B demonstrate that diagnostic palaeoscolecid characters readily preserve at Waukesha, and thus their absence in *Macromyzon* is not a function of taphonomy. (C) A specimen of *Wronascolex antiquus* from *García-Bellido, Paterson & Edgecombe (2013)*, the most convincing possible palaeoscolecid moult known due to the breakage (white arrow) at the midline; however, the authors acknowledge that this is not a surety. (D) A close-up on a piece of the scleritome of *Hadimopanella* showing interlocking articulation of the sclerites. The sclerites of palaeoscolecids in general are unlike the more labile tubercles found in leeches and *Macromyzon* in particular; photo by *Topper et al. (2010)*.

(Figs. 3A, 3B), but completely absent in *Macromyzon*. In addition, palaeoscolecids from Waukesha are often preserved with traces of the gut (see Fig. 7D in *Wendruff et al. (2020a)*), which are dissimilar to anything found in *Macromyzon*, and the overall integument of *Macromyzon* is more flexible than those found in palaeoscolecids.

The possibility that it may represent a palaeoscolecid moult is similarly unconvincing for several reasons. First, *Macromyzon siluricus* lacks features that would presumably be evident in such remains (*e.g.*, sclerites, scalids, posterior hooks, an introvert; compare with Figs. 3A–3D). Second, no palaeoscolecid moults have ever been confidently identified, but some–including the specimen shown in Fig. 3C–have been discussed in the literature (*García-Bellido, Paterson & Edgecombe, 2013*). Typically, specimens identified as palaeoscolecid moults take the form of disarticulated cuticular fragments, which is logical

for discarded exuviae (*Zhang & Pratt, 1996*; *García-Bellido, Paterson & Edgecombe, 2013*; *Daley & Drage, 2016*), but incongruous with the entirely articulated and well preserved specimen illustrated here (Fig. 1). Here, it may be informative to consider the manner in which extant priapulids–the best extant models for palaeoscolecids–undergo ecdysis. *Priapulus* moults through the formation of a longitudinal split that begins at the proboscis and gradually lengthens (*Wang et al., 2019*). Shed cuticle near the caudal appendages also wrinkles and folds as the animal emerges from its exuvia; the morphology of such exuviae would not be consistent with *Macromyzon*. Finally, our phylogenetic analyses, the results of which are presented below, included several vermiform ecdysozoans (including both extant priapulids and the palaeoscolecid *Cricocosmia*), and argue against any ecdysozoan relationship.

We have also considered the possibility that *M. siluricus* represents a fragment of another fossil but have found that argument unconvincing after careful consideration of morphology. The overall outline of the fossil is sharp and well defined without any appearance of breakages or ragged edges that might be interpreted as tearing points. It is difficult to conceive of a taphonomic setting that would result in the subdivision of a soft-bodied, vermiform animal into perfectly smooth constituent parts without any evidence of breakage, and we do not know of any such examples in similar soft-tissue-yielding Lagerstätten. There is also the fundamental issue that no currently described taxon from Waukesha–or the Silurian for that matter–has an anatomy of which *M. siluricus* could logically represent a fragment. In addition, the soft tissues of polychaetes from Waukesha have been identified, although not formally described (see Fig. 7F in *Wendruff et al., 2020a*), indicating that their annelid histology falls within the taphonomic window of the Waukesha locality. Indeed, even more taphonomically labile body plans such as non-cuticularized early chordates have been recognized from this biota (see Figs. 8F–8G in *Wendruff et al. (2020a)*).

### Phylogenetic analyses recover *Macromyzon* as a stem leech

Phylogenetic analysis resolves *Macromyzon siluricus* as sister to extant leeches (posterior probability (PP) = 0.60) within a strongly-supported clade consisting of *Macromyzon* and the three extant orders of Hirudinea (posterior probability = 1; Fig. 4A, File S2). Although the parsimony tree is less resolved, the Hirudinea + *M. siluricus* clade is still present and well-supported (bootstrap (BS) = 80%; Fig. S2, File S3).

To account for ambiguity in the interpretation of segmentation, we conducted two sets of supplementary analyses. For the first, the subdivision of segments into annuli (character 61) and the annulation pattern of midbody segmentation (character 62) are both coded as "unknown". The resolution of hirudinid taxa differs slightly in the resulting Bayesian phylogeny, with *M. siluricus* belonging to a polytomy with Acanthobdellida and Hirudinida (PP = 0.58; Fig. S3, File S4). There is no change in the parsimony topology (BS = 74%; Fig. S4, File S5). These same results are recovered for the second set of analyses, in which subdivision of segments is coded as "present" and pattern of midbody segmentation is coded as "unknown" (PP = 0.56; BS = 80%; Figs. S5–S6, Files S6, S7). In all,

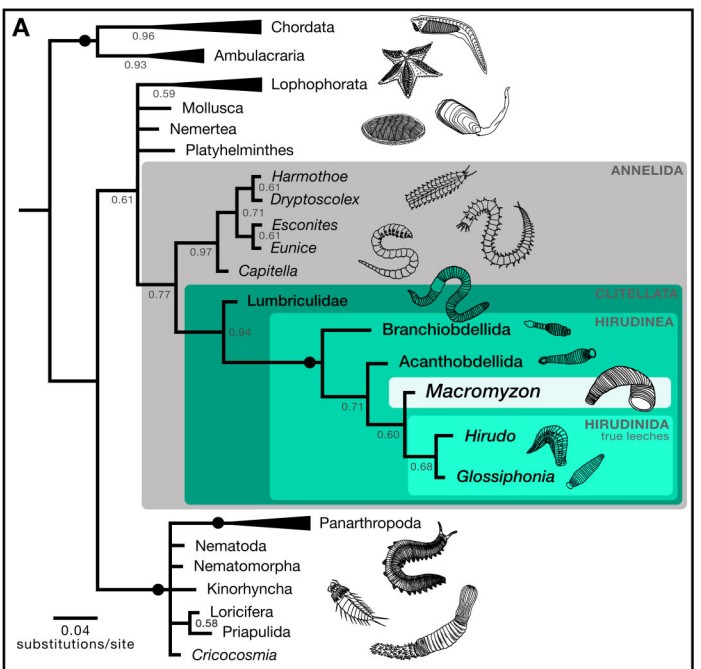

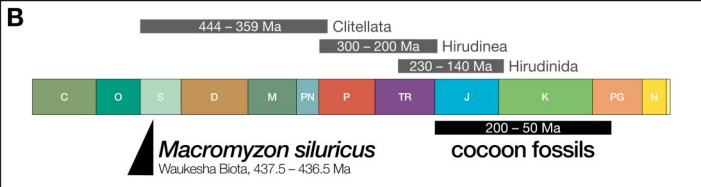

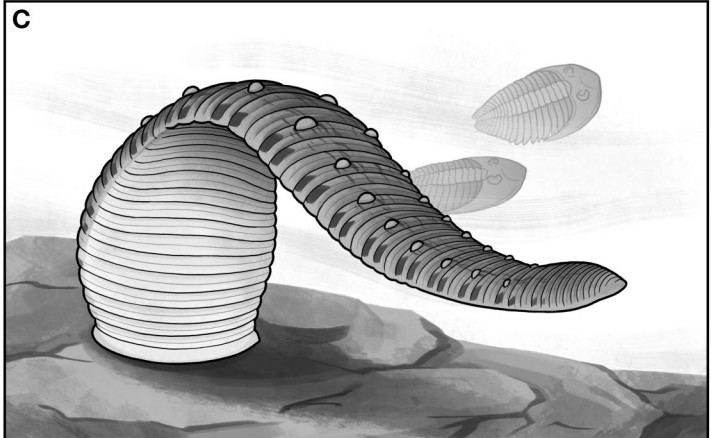

**Figure 4 Phylogenetic analyses recover *Macromyzon* as a stem leech.** (A) Bayesian inference tree: black circles on branches indicate nodes at which posterior probability (PP) = 1, other PP values are displayed below branches. Triangles represent collapsed clades with the length of the triangle corresponding to the length of the longest branch. (Uncollapsed topology is available as File S2. Maximum parsimony topology is shown in File S3). Illustrations show representative taxa for each clade. (B) Geologic timescale comparing clade age estimates from molecular clock analyses (grey bars) above, age of cocoon fossils (black bar), and age of *Macromyzon siluricus*. (C) Life reconstruction of *M. siluricus*. Illustration by E. K. Chan.

these supplementary analyses imply that the taxonomic position of *Macromyzon* is robust to alternative interpretations of annulation and segmentation pattern.

In sum, our phylogenetic analyses indicate that *M. siluricus* is, at minimum, a stem hirudinean, which predates the estimated origins of that clade by more than 130 Ma. Considering these results in combination with our morphological analysis, we argue that *Macromyzon siluricus* is most likely a stem leech.

## DISCUSSION

Morphology and phylogenetic analyses suggest that *Macromyzon siluricus* is a stem leech (Figs. 1, 2, 4C). The segmentation pattern and general body shape–including the lack of a bulbous pharyngeal region–argue against an affinity with Branchiobdellida (Fig. 2D). The caudal sucker is well-developed (Figs. 1A–1B, 2A), contrasting the modern acanthobdellidan condition in which the sucker is not clearly demarcated from the body, and has been considered rudimentary (Fig. 2E) (*Utevsky, Sokolov & Shedko, 2013*; *Bielecki et al., 2014*; *de Carle et al., 2022*). Although the lanceolate body shape of *M. siluricus* is similar to modern leeches in the family Glossiphoniidae (Fig. 2G), we are unable to address the finer-scale phylogenetic affinities of this new taxon due to the lack of internal characters and the absence of finer morphological detail in the anteriormost region of the
body. Nevertheless, *Macromyzon* has implications for aspects of clitellate evolution, including timing, habitat transitions, and the evolution of parasitism.

## Evolutionary origins of leeches

The interpretation of *Macromyzon siluricus* as a stem-group leech implies that it represents a critical data point for constraining the timing of clitellate and hirudinean origins. Three fossil-calibrated molecular clock estimates have placed the origin of Clitellata between the Silurian (Llandovery) and the Late Devonian, 444–359 Ma (*Edgecombe et al., 2011*; *Erwin et al., 2011*; *Erséus et al., 2020*). These analyses also estimate that the split between Hirudinea and oligochaetes occurred 300–200 Ma, and that true leeches appeared 230–140 Ma, but most likely in the Jurassic or Cretaceous (Fig. 4B). The presence of *M. siluricus* in the Waukesha Biota (Silurian: Llandovery, Telychian), constrained to 437.5–436.5 Ma, is consistent with earlier molecular clock estimates for the divergence of clitellates, and predates the minimum estimated origins of true leeches by more than 200 million years. Given the relatively derived position of this taxon within that class, the origins of Hirudinea and Clitellata as a whole may precede the Llandovery.

It must be noted that discrepancies between our findings and these previous studies are to be expected. First, and most significantly, because the accuracy and precision of node age estimates improve as the number of fossil calibrations increases (*Warnock, Yang & Donoghue, 2017*), we should not expect accurate or precise age estimates for lineages–such as clitellates–with sparse geologic records. Second, none of the three studies was principally interested in the origins of Hirudinea. Two focus on the origins of Metazoa broadly, and the nearest fossil calibration points are phylogenetically distant from Clitellata: being at the base of Annelida (*Edgecombe et al., 2011*) or within Mollusca (*Erwin et al., 2011*). The third study, focused on Clitellata as a whole (*Erséus et al., 2020*), has only a single clitellate fossil at their disposal: 201 Ma leech cocoons (as described in *Manum, Bose & Sawyer (1991)*). The authors conservatively used these cocoons as the minimum age calibration point for the divergence between Hirudinida and Branchiobdellida; however, the original descriptors (*Manum, Bose & Sawyer, 1991*) and subsequent authors (*e.g.*, *Bomfleur et al., 2012*) remarked that the ultrastructure of the oldest fossil cocoons is markedly similar to those produced by more derived true leeches within the suborder Hirudiniformes. This indicates that they could be suitable for use as a calibration point for shallower nodes, and helps explain why these Triassic fossils may underestimate the true age of Hirudinida. The other fossil calibration point used by *Erséus et al. (2020)* is *Coprinoscolex*, an echiuran from Mazon Creek (307 Ma). Echiura is a relatively distant relative of Clitellata (*Weigert & Bleidorn, 2016*) and the fossil is much younger than other known annelid material. It is plausible that the inclusion of a relatively recent fossil at such a distant outgroup could have favoured younger estimates for ingroup nodes. Without additional fossil discoveries, however, molecular clock age estimates for these clades cannot be improved.

At least three scenarios may explain the paucity of described leech fossils. These hypotheses are not necessarily mutually exclusive. (1) Leeches may have originated early in the Paleozoic, but remained at low diversity and abundance for most of their evolutionary

history, reducing their likelihood of fossilization. (2) Localities in which leeches thrived or diversified were not conducive to exceptional fossil preservation: this hypothesis is consistent with the proposed diversification of some leech lineages–and clitellates in general–in freshwater ecosystems. (3) For a variety of reasons, crucial diagnostic characters are not visible or present in collected specimens, precluding their identification as hirudineans.

The last two scenarios, in particular, bear additional discussion, and help contextualize the disparity between the age of this new taxon and previous leech fossils. By virtue of their almost entirely soft-tissue anatomy, leeches are unlikely to fossilize except under the most exceptional conditions such as those found in the Burgess Shale (*Nanglu, Caron & Gaines, 2020*), Mazon Creek (*Clements, Purnell & Gabbott, 2019*), and other Lagerstätten (summarized in *Babcock (2025)*). As a result, the temporal window within which we might reasonably expect to recover a fossil leech is not only narrow, but heterogeneously distributed through time and space (*Van Roy, Briggs & Gaines, 2015*). Thus, the large temporal discrepancy between the heretofore only convincing traces of leeches in the fossil record (*i.e.*, cocoons) and *Macromyzon* in the Silurian is not surprising from a stratigraphic perspective. Most stratigraphically intermediate taxa were unlikely to be captured in the fossil record due to the relative lack of taphonomically appropriate sites. This is particularly true in light of arguments that most clitellates, including many hirudinean lineages, are presumed to have diversified in freshwater environments (*Rousset et al., 2008*; *Erséus et al., 2020*), which are less conducive to the preservation of soft tissues.

The soft-bodied nature of leeches also may contribute to the difficulty of recognizing fossil taxa as definitive leeches, even when they are preserved. Incomplete preservation of original morphology is common among fossils, doubly so for soft-tissue structures (*Sansom, Gabbott & Purnell, 2010*; *Nanglu, Caron & Cameron, 2015*; *Babcock, 2025*). Thus, leech body fossils may not always be able to be easily recognizable if critical features are obscured or lost through taphonomic processes, as might be the case with disarticulation or decay. Investigation of newly discovered Lagerstätten or reinvestigation of known localities may help bridge the apparent discordance between the accepted stratigraphic ranges and this new discovery. Mazon Creek, in particular, is a promising area to search for additional clitellate fossils. Mazon Creek preserves a mixture of freshwater and marine species, and has a high fidelity of soft-tissue preservation (*Clements, Purnell & Gabbott, 2019*). It is Carboniferous in age, temporally intermediate between the Triassic cocoons and the Silurian *Macromyzon*. In addition, new species continue to either be described (*Mann & Gee, 2019*) or redescribed (*McCoy et al., 2016*) from the site despite its long history of publication, suggesting a degree of cryptic diversity that may include additional annelids.

## Habitat

The most recent clitellate ancestor was an aquatic organism. The common consensus is that the earliest clitellates inhabited freshwater environments, with some lineages secondarily becoming marine (*Rousset et al., 2008*; *Erséus et al., 2020*). Clitellate cocoons have been interpreted as a necessary prerequisite for freshwater and terrestrial life, as they

protect eggs in hostile, non-isotonic environments (*Manum, Bose & Sawyer, 1991*; *Westheide, 1997*). Additionally, most modern aquatic leeches live in freshwater, with the exception of the principally marine suborder Oceanobdelliformes. The marine palaeoenvironmental setting in which *Macromyzon* was discovered gives cause for re-examining these long held assumptions.

If *Macromyzon* is indeed a stem leech, there are three possible ways to explain the distribution of marine taxa within Hirudinea. First, *Macromyzon* and the hirudinidan clade Oceanobdelliformes–which is sister to all other leeches (*Trontelj, Sket & Steinbrück, 1999*; *Tessler et al., 2018*)–may each represent independent transitions from freshwater to marine environments. Second, it is possible that a freshwater-to-marine transition occurred on the branch leading to Hirudinida, with a subsequent recolonisation of freshwater in the clade encompassing the non-oceanobdelliform true leeches. The third, and most plausible, possibility is that the ancestor of Hirudinea was marine, and that the non-leech, freshwater orders Branchiobdellida and Acanthobdellida represent independent transitions to freshwater. Both of these orders are separated from other taxa by long branches (*Tessler et al., 2018*; *de Carle et al., 2022*). For the branchiobdellidans, this implies a recent and rapid radiation; the Acanthobdellida is incredibly species-poor, with only two known extant species. Acanthobdellidans are primarily parasites of salmonid fishes, and branchiobdellidans are obligately commensal on astacoid crayfish (*Skelton et al., 2013*; *Utevsky, Sokolov & Shedko, 2013*; *Bielecki et al., 2014*; *de Carle et al., 2022*). Both hosts stem from marine lineages; therefore, these taxa may have accompanied their hosts on their freshwater transitions whereas the earliest leeches remained marine.

## Parasitism

Leeches are unique among annelids as a radiation of blood-feeding parasites. Although not directly used in feeding, it is tempting to speculate that the posterior sucker, which defines Hirudinea, could be an adaptation to parasitism due to its use in host-attachment; yet, some aspects of hirudinean ecology cast doubt on this speculation. Many extant leeches are predatory, rather than parasitic. These taxa use their suckers to manipulate prey or to attach to substrates (*Sawyer, 1986*). Furthermore, branchiobdellidans, which have tight associations with their crayfish hosts, are broadly ectocommensal rather than parasitic (*Skelton et al., 2013*); however, evidence suggests that their "suckers" are homoplastic with those of Hirudinida and Acanthobdellida (*Weigl, 1994*).

Other features–the subdivision of annuli, elaboration of the crop, and lack of septa in mid-body segments–have been postulated to facilitate sanguivory in leeches by allowing them to expand to store copious quantities of blood (*Sawyer, 1986*). Because these, in addition to several anticoagulants potentially implicated in blood-feeding (*Iwama et al., 2019*, *2021*; *Iwama, Tessler & Kvist, 2022*), are present in all true leeches regardless of feeding mode, most authors have postulated that the ancestral hirudinidan fed on vertebrate blood (*Trontelj, Sket & Steinbrück, 1999*; *Tessler et al., 2018*)—rather than parasitizing invertebrates or engulfing them whole as many extant leeches do (*Sawyer, 1986*)—and that sanguivory is likely a synapomorphy of Hirudinida. The composition of

the Waukesha Biota and the age of *Macromyzon*, however, call this hypothesis, and the distinction between vertebrate and invertebrate parasitism, into question.

No crown vertebrates have been recovered from the Waukesha Lagerstätte. Chordate remains, such as conodonts and other undetermined chordates, are present in the Waukesha Biota (*Wendruff et al., 2020a*; *Randolphe & Gass, 2024*), but are extremely rare, and were likely too small to serve as viable hosts. Based on the overall faunal composition of the Waukesha Biota, it seems implausible that *M. siluricus* fed on vertebrate blood. *Macromyzon* most likely fed on invertebrates, either as a predator or as a parasite.

The most abundant animals at the Waukesha locality are arthropods, with trilobites being the most numerous of these (*Mikulic & Kluessendorf, 1999*; *Wendruff et al., 2020a*). Given that some modern leeches obligately or facultatively feed on arthropod haemolymph and other bodily tissues (*Sawyer, 1986*; *Nakano et al., 2017*), trilobites and other arthropods represent the most plausible hosts for *M. siluricus*. Interestingly, certain extant hirudineans also have non-parasitic associations with arthropods: some marine leeches deposit their cocoons on the hard shells of arthropods and bivalves, and branchiobdellidans live exclusively on freshwater crustaceans (*Sawyer, 1986*; *Skelton et al., 2013*). It is therefore possible that these associations, both parasitic and commensal, share a common origin.

## ACKNOWLEDGEMENTS

We thank E.K. Chan for enthusiastically providing the life reconstruction, D. Currie for taxonomic advice, and S. Kvist and C. Moreau for comments on an earlier version of the manuscript. We are grateful to D. García-Bellido, S. R. Gelder, A. S. Y. Mackie, T. Nakano, J.-F. Parpet, J. R. Patterson, P. Świątek, T. P. Topper, A. Utevksy, and S. Utevsky for generously sharing photos of leeches, crayfish worms, hook-faced fish worms, and palaeoscolecids. We thank H. Drage and the anonymous reviewers for their thoughtful commentaries on the manuscript. The late G. O. Gunderson and R. C. Meyer collected this specimen, and we thank C. Eaton for arranging its loan.

### Funding

This work was supported by a Natural Sciences and Engineering Research Council of Canada Postgraduate Doctoral Scholarship (No. PGSD2-518435-2018) awarded to Danielle de Carle, and a Smithsonian Institution's National Museum of Natural History Peter Buck Deep Time Postdoctoral Fellowship awarded to Karma Nanglu. The funders had no role in study design, data collection and analysis, decision to publish, or preparation of the manuscript.

### Grant Disclosures

The following grant information was disclosed by the authors:
Natural Sciences and Engineering Research Council of Canada Postgraduate Doctoral Scholarship: PGSD2-518435-2018.

Smithsonian Institution's National Museum of Natural History Peter Buck Deep Time
Postdoctoral Fellowship.

## Competing Interests

The authors declare that they have no competing interests.

## Author Contributions

- Danielle de Carle conceived and designed the experiments, performed the experiments, analyzed the data, prepared figures and/or tables, authored or reviewed drafts of the article, and approved the final draft.
- Rafael Eiji Iwama conceived and designed the experiments, performed the experiments, analyzed the data, prepared figures and/or tables, authored or reviewed drafts of the article, and approved the final draft.
- Andrew J Wendruff conceived and designed the experiments, performed the experiments, analyzed the data, prepared figures and/or tables, authored or reviewed drafts of the article, and approved the final draft.
- Loren E Babcock conceived and designed the experiments, performed the experiments, authored or reviewed drafts of the article, and approved the final draft.
- Karma Nanglu conceived and designed the experiments, performed the experiments, analyzed the data, prepared figures and/or tables, authored or reviewed drafts of the article, and approved the final draft.

## Data Availability

Raw data is available in the Supplemental Files.

## New Species Registration

The following information was supplied regarding the registration of a newly described species:

Publication LSID: urn:lsid:zoobank.org:pub:D5B80E8C-F0F8-4FDC-A617-29E31F6745AE

Macromyzon: urn:lsid:zoobank.org:act:56E87D9F-D838-4FFA-AC47-57825B46EAED

Macromyzon siluricus: urn:lsid:zoobank.org:act:8F3D2E52-2AE0-4F6C-9C8B-0AE1C62EA423

## Supplemental Information

Supplemental information for this article can be found online at http://dx.doi.org/10.7717/peerj.19962#supplemental-information.

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
