# Peer review of "The first leech body fossil predates estimated hirudinidan origins by 200 million years"

_PeerJ, doi:10.7717/peerj.19962_

## Round 0.1 · original submission · Major Revisions

Please, address all the concerns that the reviewers raised. I found the issue about the systematic placement and validity of phylogenetic analyses raised by Reviewers 1 and 2 critically important for accepting the manuscript.

Reviewer 1 ·

Basic reporting

Experimental design

The principal aspects of the description are inadequately justified by the fossil material illustrated.

The phylogenetic analyses are based on over-interpretation of morphological information and do not incorporate the extremely strong stratigraphic evidence that contradicts the authors' main interpretation.

Validity of the findings

- The presence of segmentation is not adequately demonstrated
- The presence of a caudal sucker is not demonstrated
- The possibility of convergence is not sufficiently considered
- There is an insufficient attempt to explain the striking discordance between the leech interpretation and the stratigraphic record, and to demonstrate that this is supported by the available evidence.

Additional comments

de Carle et al. present a new exceptionally preserved fossil from the Silurian Brandon Bridge Lagerstatte, which they interpret as a hirudinean leech. This would have the consequence of pushing back the origin of Clitellata (a major annelid clade) by hundreds of millions of years, implying that a major but cryptic radiation of non-marine annelids had taken place by the early Silurian. The question is: is the interpretation of the single, imperfectly preserved specimen sufficiently unambiguous to support an unquestionably major finding that would prompt a broad-brush reconsideration of annelid evolution?

The authors' argument can perhaps be condensed into two strands: firstly, that the annulations and a caudal sucker are sufficiently similar to those of hirudean leeches that they must be homologous; secondly, that no other affinity can parsimoniously account for the observed morphology.

In my view, the description is insufficiently detailed to support the first contention, and is not supported by the fossil material. On the second point, insufficient data are available to rule out the possibility that the specimen is a derived member of another lineage. The fossil is certainly intriguing, and a careful, conservative description would merit publication. But a measured approach is necessary; the claim that this single fossil with ambiguous morphology provides sufficient evidence to overturn everything we thought we knew about leech evolution is overstated.

The more detailed comments below are adapted from similar comments made on previous versions of this manuscript.


## Description

Major aspects of the description are inadequately justified.


How are the dorsal and ventral surfaces identified as such?

I can see no basis for defining segments based on groupings of six annulae; the brackets in Fic. 1C do not seem to correspond to any clear grouping that I can see, and the “segment” boundaries are not drawn in Fig. 1D. Whilst the organism was clearly annulated, I do not see any basis for recognizing segments. The positions of the tubercles indicated in the schematic 1E do not correspond to the positions drawn on 1D, which do not seem to correspond to genuine biological features on the specimen itself: the arrowheads on 1C seem to point to a subset of ?taphonomic flecks on the fossil surface that are not obviously biological, and that exhibit no obvious differences from a large number of similar flecks that are neither arrowed in 1C nor depicted in 1D.

I do not understand how the authors interpret the prominent change of width. The reconstruction sketched in Fig. 2C implies that the width of the organism gradually increased from posterior to anterior, but that does not match with the morphology I see in the fossil, in which the posterior section has an almost constant width, which is about half the width of the anterior section. I do not see how twisting or folding of a lanceolate body outline would lead to two such consistent and distinct body widths, particularly given the absence of wrinkling or deformation on any part of the trunk apart from the join of the anterior and posterior sections.

How is the absence of chitin in the cuticle (line 237) established?


## Potential leech homologies

I have difficulty seeing any detailed similarity between the open ring of Macromyzon and the closed Hirudinea caudal sucker. I cannot see any indication of a change in trunk width that characterizes the modern suckers figured in fig 2. The two dark concentric rings in the fossil material seem not to have any obvious parallel in extant leeches. A specific list of detailed similarities that the fossil is meant to share with the caudal suckers of specific extant taxa would be helpful, as I do not see any resemblance between the figured caudal suckers and the dark peripheral ring in the fossil (which does not have an obvious biological interpretation under the leech interpretation).

With no convincing evidence of segmentation, or of a caudal sucker, I see no reason to attribute this fossil to a taxon that is not thought to have evolved until many millions of years later. And indeed, when push comes to shove, the organism is very short on morphological characteristics. I think it is an error to pretend that its affinity can be established with any confidence; any similarities to modern groups (if genuine) are sufficiently non-specific that they might readily be attributed to convergence. Given these interpretative uncertainties, and the fundamental lack of morphological characters available in the fossil, running a phylogenetic analysis on such a poorly known taxon is a pointless exercise; if the taxon is coded as exhibiting leech synapomorphies, it is somewhat inevitable that it will come out as a leech.

If the authors put any faith in molecular clock results and the fossil calibrations on which they are founded – and it sounds like they do – then they will appreciate that these lines of evidence place an extremely strong prior on Macromyzon not belonging to a group that did not evolve until the Permian. If the authors are really set on running a phylogenetic analysis, they should select a model that incorporates this stratigraphic information. A tip-dated model could in principle test whether the available morphological evidence provides strong enough evidence to override the available stratigraphic information. The authors’ interpretation violates the hard upper bounds used in previous molecular clock analyses; the authors must discuss why these bounds are invalid and how their interpretation can therefore be reconciled with the fossil record.


## Alternative affinities

I can’t understand why the authors devote so much ink to discounting a palaeoscolecid affinity – which can be readily dismissed with the simple observation that palaeoscolecids have a cylindrical trunk of constant diameter (not to mention an eversible proboscis, scalids, and posterior tail hooks).

The pronounced change in body width associated with a twist in the specimen does not have any obvious counterpart in leeches, but does somewhat resemble the condition in vetulicolians such as Banffia (Caron 2005/6, Trans Roy Soc Edinburgh 96). A reasonably strong case for a vetulicolian affinity could be built on: a bipartite trunk with a broad 'anterior' separated from a narrower annulated 'posterior' by a twist; an inner and outer robust terminal circlet surrounding a 'mouth'; and ridges close to the location of the twist (compare Fig. 1C with e.g. Caron fig. 8). A similar body outline can be found in Yuyuanozoon and Heteromorphus (see e.g. Chapter 26 in Hou et al. 2017, “ The Cambrian Fossils of Chengjiang, China: The Flowering of Early Animal Life”). The apparently blunt termination of the posterior end seems to recall the condition in certain vetulicolians more closely than in hirudeans.

I’m not particularly convinced by this argument, even if I find it more compelling than the leech interpretation. But if the authors are going to propose radical interpretations of the fossil, they may as well be exhaustive in the possibilities they consider.

My strong recommendation is that the authors acknowledge that the new fossil is incertae sedis. A careful and complete description on this basis would pave the way for further finds that eventually illuminate its evolutionary significance.

Reviewer 2 ·

Basic reporting

For the most part, the language used is professional, scholarly English, and is clear and easy to understand. A few exceptions that I found:

Line 44: This would be a good place to redefine what the Clitellata are; I know this was included a short while ago in the abstract, but the abstract and introduction should be thought of as separate entities.

Line 65: “huge” is a little informal and maybe a bit of an overstatement. “Broad” or “Large” would be better word choices

Line 67 to 69: I would move the sentence starting with “With the…” to the end of the previous paragraph, which defines the synapomorphies of the Clitellata and the taxa contained within it.

Lines 75 to 95: These two paragraphs have good ideas, but the focus of the paragraphs jumps around a bit. I think a more logical order would be to compress the sentences from lines 85 to 95 into a single paragraph (“Exceptionally preserved leech cocoons…” to “…and the origins of parasitism”) and then move it to after the first sentence of the paragraph starting at line 75. This way, you remark about the known clitellate fossil record, note the problems with it's scarcity, and then discuss what, prior to this, was maybe an alleviation as to that scarcity (but probably not).

Line 101: The way this sentence is structures, it makes “leech” sound like a tissue, not an animal with labile tissues. Could be made to be less ambiguous.

Line 151: “groups” sounded a little ambiguous to me. I would say “genera (and the respective groups to which they belong)” there.

Line 152: replace “the orders” at the end of the line with “four taxa comprising Hirudinea:”. Then, on line 154, you can erase the sentence “Together, these four taxa comprise the subclass Hirudinea.” This will help streamline the paragraph.

Lines 193-194: This sentence on the paleoenvironmental context feel a little tacked on in the stratigraphic occurrence section, and beside that was not included in a supplemental file (more on that below).

Line 217: Would vermiform-lanceolate be a standard designation for leech body shapes? I would almost call it pyriform-lanceolate, but I am not as well-versed in leeches as the primary author, so I defer to their ultimate judgment.

Line 218: I would say “anterior” rather than “forward,” to keep the terminology technical

Line 229: I think you meant “…then narrowing slightly toward the posterior margin” at the sentence’s end.

Line 237: I’m not entirely sure how to fix this, but you wouldn’t actually be able to see or determine “chitinous cuticle” in a fossil with the level of analysis you’ve done. Maybe mention the lack of jointed or lobopodous limbs to exclude the fossil from the panarthropods?

Line 259: Note the subfigures for the branchiobdellidans and acanthobdellidans in Figure 2.

Line 270: In Figure 1, the “metameric circular organs” are simply called tubercles. I was pretty sure I knew what you were talking about, but consistent terminology would ease the reading.

Lines 282-283: It is not immediately obvious to me why the number of segments is enough to confidently say there is no external clitellum. If the taphonomy is good enough for segments it should be good enough for a clitellum? Is it that it would be anterior on the body?

Line 285: there is an extra “and” within the parentheses.

Line 301: The whole paragraph details multiple reasons why a palaeosolecid molt is an unlikely interpretation. Using a colon implies you can list them all in that sentence. I would end the sentence starting on line 301 by simply saying “…is similarly unconvincing for multiple reasons.” Then picking up with “M. siluricus lacks features that…”

Line 303: after the parentheses, rather than say “and no,” I would write “although admittedly”. The fact that we don’t have convincing palaeoscolecid molts does not help your contention that M. siluricus is not a paleoscolecid molt. Absence of evidence is not necessarily evidence of absence.

Line 308: “it may also behoove us” sounds kind of informal. For this line, I would start “Here, consideration of the molting behavior of priapulids–the best extant model for paleoscolecids–may be informative.”

In Figure 1’s caption, Tu is not defined on it's own, only as Tu?, but Tu without a ? may be found on subfigures E and G.


The literature references are quite thoroughly researched for matters relating to leeches and Clitellata, but they could be more rigorously cited when it comes to the taxa of the Waukesha biota. Wendruff et al. 2020a is relied on quite heavily when discussing the taxonomic and paleoenvironmental background. Mikulic et al. 1985a and b should also be included more often, as should Mikulic and Kluessendorf, 1999 when discussing the stratigraphic context of the Brandon Bridge Formation, and Kulessendorf 1990 is a good, if older, reference for the Waukesha lagerstätte generally. This dissertation was the first thorough treatment of the lagerstätte.

Within the works cited itself, there are a few errors I found:
Line 453: Paracanthobdella livanowi should probably be italicized.

Lines 516–517: This is an incomplete citation, and an important one at that! Make sure you fix it

Lines 629¬–631: I could not find this citation anywhere in the paper, although I think I saw where it was meant to go when discussing recently described Waukesha taxa. If you do keep it, you should include papers like Pulsipher et al. 2022, Anderson et al. 2021, and Jones et al. 2015, too.


The structure of the paper itself is appropriate and easy to follow, and the information for the main portion of the paper appears to be complete. There is some information that appears to be either short or missing from the Supplemental Files, though. Lines 160 to 161 mention that the new characters and their coding are described in the supplementary information. While technically true, I was hoping for a bit more of a discussion as to why they were included and their significance, rather than a statement on just what they were. On lines 193 and 194, it says that the paleoenvironmental context of the Brandon Bridge Formation is in the supplemental files, but this appears not to be included at all. For supplements 2 and 3, I would encourage including a jpeg or tif file of the tree for ease of access. It's quite easy to convert .tre files to a more accessible visual formats using a program like FigTree.
Finally, one small thing I noticed is that in Figure 1, subfigure A is not labeled beforehand as (A).

The article is nicely self-contained, and does not leave any major loose ends, so to speak. Some areas of the discussion could have been developed more, but I will note those later. At the very end of the introduction, I would make it a little more explicit about what the study will do in terms of phylogenetic analyses and morphological comparisons to other phyla it could potentially be a part of. This wouldn’t take much extra text, as you already describe how you will demonstrate that it is a hirudinean taxon. I say this because the mention of the inclusion of Cricocosmia in your phylogenetic matrix in lines 154–156, while certainly a good idea, was done to test a hypothesis you had not detailed before.

Experimental design

This article certainly falls within the Aims and Scope of PeerJ’s mission, as it is a research article in the (historical) biological sciences. Although, as detailed previously, the elucidation of the hypothesis could be a little clearer in the Introduction, the relevancy and meaningfulness of the research is quite obvious: the described taxon can reasonably be seen to be the earliest known representative of the hirudineans, potentially by an awful long time, and is an important fossil for a rarely represented group. While the methods described are done so in sufficient detail to understand what is happening and to replicate for further research, it is with the execution of the methodology that the article runs into it's greatest problems.

First, I do want to note that I think that there is a very good chance that the included fossil is a hirudinean of some sort or another. At the Waukesha, it is not unusual for soft-bodied kerogenized tissue to look darker when it is folded on top of itself (as is seen when the specimen twists midway along it's length), but the feature at the tail end of the fossil seems to not merely be folded over itself, but to obscure the annulations behind it, which is what would be expected when comparing the fossil to the smooth suckers seen in the living representatives. The segments are generally not strongly visible (what I thought were segment boundaries near the upper left of the worm in Fig. 1C are an annulus or two off of where they were marked), but the two-tiers of annulations are easily visible and intriguing. My issue with the methodology thus does not lie with the morphological but phylogenetic analysis.

The big headline of the article is that the Hirudinea must have evolved several hundred million years before conventional estimates suggest they did since the new species, Macromyzon siluricus is placed nested within the group, closer to the crown Hirudinida than the living (and perhaps relatively recently evolved?) Branchiobdellida and Acanthobdellida. Yet, the complete lack of resolution of Ecdysozoa and Lophotrochozoa outside the Annelida prompts an examination of the character table used.

I do want to commend the work that had to go in to add many new taxa and characters to the table of Ou et al., 2017 (and deleting the invariant ones) as I know how time consuming this process is, but the resulting phylogenetic matrix is lopsided, with some characters meant to make basic differentiations between phyla and subphyla, and others that are better suited for finer-scale differences between genera. Further, multiple taxa were excluded, and it was not stated that their inclusion would not make an impact on the structure of the trees, which is the only situation I could see justifying their exclusion. To say that there is no apparent connection between those taxa and M. siluricus is not really true, as you include other taxa, like deuterostome phyla, that don’t seem to have any obvious connection to the new taxon.

In short, I suspect that the results of the phylogenetic analysis are not entirely reliable, and they are not discussed very much besides, only from lines 331 to 337 in the text. Here is what I would suggest: keeping both a Bayesian and a Maximum Parsimony analysis, run two separate phylogenetic analyses, the first a broad, phylum-level analysis with the new taxon and another annelid; then run a second, more specific analysis to narrow down the placement of M. siluricus within the Annelida. This second analysis may be rather character-poor with your current table, so you may have to add some more (perhaps including details of the tubercles?) but it will also have fewer taxa to begin with, so you will have that in your favor. I am wondering if M. siluricus might not fall towards the base of the hirudinean tree in this scenario; you make a good point in saying that the divergence time calibrations of Edgecombe et al. 2011 and Erwin et al. 2011 were not focused on the Clitellata, but I would be surprised if the calibrations of Erséus et al. 2020 were that far off. But maybe I’m wrong! A more rigorous phylogenetic analysis could answer it.

Validity of the findings

Impact and Novelty were not explicitly assessed, as is appropriate for the journal. The underlying data have been provided and are robust and sound, excepting the objections I went over in the previous section. The conclusions of the paper are generally well-written and connect back to the earlier sections of the paper and the data well. I particularly enjoyed the “Habitat” and “Parasitism” subsections of the discussion. Notwithstanding the ultimate taxonomic position of M. siluricus, the idea that leeches might have started as parasites or commensals on arthropods is interesting, and the non-parsimonious explanation of marine adaptations in Hirudinea makes sense in the “Habitat” section. There are a few points that I thought could have been explored a bit more, although I generally liked the sections they were in, too. First, the discussion of the placement of the “?leech” in lines 77–85 is a good idea to have, as I thought the leech being described was that specimen at first. I would be interested in more insight on why what superficially appears to be a leech probably isn’t one in the Discussion.

Second, the “metameric circular organs” or tubercles are interesting, and would be good candidates for additional characters in your phylogenetic matrix. I’m not sure how, even looking at the high-resolution images provided with the manuscript, you can distinguish between a tubercle and a blotch of taphonomic variability on the surface of the fossil. Do they show slight topographic relief? If so, a subfigure with a slanted light angle could be good to add to the figures. Third, when discussing how the fossil’s preservation is incongruent with a paleoscolecid (among many other taxa), it's also worth mentioning that, as you can see from previous publications, paleoscolecids in the Waukesha often preserve intestinal tracts (having worked on the fauna myself, I can confirm this), and that based on the shape of the preserved specimen, the integument of M. siluricus appears durable yet soft and flexible, unlike paleoscolecids.

Additional comments

Having examined the new species policy for PeerJ, I can say that the description provided meets the ICZN standard laid out in the authors’ guide, and is appropriate and sufficient for scientifically describing the species. I also agree that the taxon is a new species and is sufficiently well preserved for it's designation. My only hesitation is that the specimen does not have an actual specimen number, but rather just an accession number. It is imperative that the physical specimen have a unique identifier for it's permanent repository at the University of Wisconsin Geology Museum.
As an additional comment I wasn’t sure where to fit in elsewhere… the subsection in the discussion “Evolutionary Origin of Leeches” feels like it could have a more integrated transition to the next subsection. As written, the numbered list appears to end a little abruptly.

·

Basic reporting

• The paper is nicely written and well explained.
• Prevec et al. (2022) Communications Biology on the middle Permian Karoo Basin Lagerstätte of South Africa should be discussed. This presents the otherwise stratigraphically oldest published supposed clitellate.

• Figure 1: picky comment, but would suggest replacing white arrow in 1F with the same label line for the caudal sucker as in A, B and G.
I will say the only issue for me is the photos not being suitably large/high quality in the review version for me to really see the putative tubercles, but these might just be too difficult to capture properly in a photo.
• From my interpretation, the putative tubercles (so called in the figure captions) correspond with what is referred to in the text as 'metameric circular organs’. I would prefer some consistency in the description of these structures, so it is easier to relate the figures to the text.
• Other figure comments: the scale bars are labelled in Figure 2 and 3 and not in Figure 1. Add the scale labels to all figures, or remove from all figures.
The figure text size is very inconsistent (the text is much larger in Figure 3), which should be resolved.
The spacing between images within each figure is also highly variable, being very close together in Figure 3, and very far apart in Figure 1. Aesthetically, I would suggest making these more uniform across the figures.
Figure 4: Cricocosmia should be italicised.

• Supplementary files: the text seems to suggest that there should be a supplementary file on the palaeoenvironmental context (L193-4).

Experimental design

No comment

Validity of the findings

• Molecular clock estimates for the origin of Clitellata are noted in the Discussion, but I think the Introduction could benefit from a sentence on the suggested divergence times of clitellates and hirudineans, because this timing (but lack of fossil evidence) is crucial to the overall significance of the paper.
• I do not think the specimen represents a palaeoscolecid moult, and I think the authors discuss this point sufficiently (e.g., the completeness of the specimen, no sclerite evidence, etc.). If it were a palaeoscolecid or priapulid (moulted or otherwise), one would also expect to see evidence of an introvert/proboscis, which there is not (the sucker does not look similar to this) – the authors might wish to also mention the lack of this character.
All this to say, I think the identification of the authors to be convincing, and it unlikely that this specimen belongs to another vermiform or ecydsozoan group (and really seems too complete to be a fragment!).
• Does the apparent low number of segments disagree with the conclusion that the species is likely a true leech (Hirudinida, 33-34 segments)? Compared to the Branchiobdellida (with 15)? This would be nice to note in the Discussion, as you mentioned the dissimilarity with the acanthobdellidan sucker, but not necessarily other within-Hirudinea potential affinities. Other aspects could also disagree with the species assignment as a true leech compared to stem hirudinean (e.g., marine environment, molecular clock estimates) – some of these could be (are almost certainly) wrong as you discuss, but worth noting. For reasons such as these, it is unclear to me why you settle as strongly on the conclusion of the species being a true leech as you do – if it is purely the phylogenetic results, I would like to have these points briefly discussed as part of making that conclusion.
• In the Parasitism section of the Discussion, I would suggest also explicitly noting that the species could have not been parasitic, based on the consideration of the biota of Waukesha?

Additional comments

Thanks to the authors and editor for sending me this paper to review.
I am not an expert on Annelida, but I am happy to provide my thoughts on the arguments and conclusions made in this paper, and to discuss the fossil record of invertebrates and ecdysozoan moults. I am convinced by their remarks on the morphological features, and the existence of a terminal caudal sucker, annulations, etc. are clear. I agree with their arguments that the fossil does not belong to Ecdysozoa, nor is it a polychaete, and generally find their points convincing.
I have a few comments, but all are very minor.
The paper is very well written, and the figures are nice and suitable as evidence. I also found the discussion of early environment and feeding mode in leeches to be very interesting, and overall enjoyed reading and reviewing the paper!
All the best,
Dr Harriet B. Drage

---

## Round 0.2 · Major Revisions

One of the reviewers raised a serious question about the actual affinity of the fossil. Please, address this thoroughly in your revised version.

Reviewer 1 ·

Basic reporting

Experimental design

Validity of the findings

As the authors emphasize in their title and abstract, interpreting this fossil as a leech requires us to rethink everything that we thought we knew about leech evolution: the timing of their origins, a marine rather than freshwater setting, the affinity of their hosts.

The authors do a thorough job of providing plausible explanations for each of these inferences (except perhaps for the absence of leeches in the well-studied Mazon Creek deposit) – but extraordinary claims require extraordinary evidence. Given that descriptions of single specimens from lagerstatten have a rich history of spectacularly missing the mark, the single available specimen simply doesn't provide the unambiguous evidence that the authors assert: it is certainly premature to base such far-reaching implications on one two-dimensionally preserved specimen that is not even sufficient to unambiguously reconstruct the three-dimensional shape of the original organism.

The authors' interpretation of the fossil material is not impossible – but then, a single specimen makes it impossible to establish with confidence whether the 'twist' is a taphonomic or biological feature; whether the 'sucker' really is a distinct structure rather than a taphonomic feature (breakage / wrinkling); whether the tubercles are real or not; whether the change in width corresponds to a change in the width of the organism or a change in the angle of burial, whether or not the animal is segmented...
Neither do I consider the taphonomic condition of the specimen to be well-constrained; the possibility of breakage or folding at either end is difficult to evaluate from just one specimen; and whilst I agree that the difference in preservation from arthropod cuticle may show that the cuticle was not as robust as arthropod cuticle, this could reflect its thickness rather than its chemical composition.

Without more specimens, ultimately all we can be confident about is that we have a broadly tubular organism that may or may not change in width and has a ring-like structure at one end; the latter may represent a distinct disc or the opening of a cylindrical tube, assuming we can really discount taphonomic breakage. Any interpretation beyond this is necessarily subjective, as there are too few specimens to fully gauge the specific taphonomic factors (twisting, folding, decay) that characterize this particular individual. The well-constrained aspects of the morphology are so few that with a little imagination they can in principle be reconciled with any number of body plans, particularly given the propensity of parasites to secondarily simplify. The paucity of unambiguous characters means that quantitative phylogenetic analysis is redundant – if the organism is coded up as a leech, it's not clear that phylogenetic analyses have the statistical power to test this hypothesis. Whilst I can see why the authors think to compare the specimen with leeches, the specimens they figure simply don't look very much at all like the fossil material, and neither do I see a plausible taphonomic process that would cause them to do so. Again, this becomes a subjective opinion due to lack of evidence: we simply don't have a rich leech fossil record against which to test different intuitions of how a leech might look once fossilized.

Whilst I appreciate the careful attention that the authors have paid to my previous comments, the bottom line is that the fossil material is insufficient to support the claims that the authors wish to make. The subjective element of taphonomic and morphological interpretation can only be reduced through the description of additional fossil material. As such, it is inappropriate to describe the specimen as an "unambiguous" leech body fossil, and premature to rewrite leech evolution on this basis. Whilst it is fruitful to probe the uncertainties in our knowledge of leech evolution, I can see no benefit in overselling the contribution that this specimen can make. Adopting a conservative approach that properly acknowledges the uncertainty around the fossil material and its interpretation would lead to a much more convincing study.

Reviewer 2 ·

Basic reporting

As before, the writing style is in clear, professional English, and has even been improved since the initial submission. Colloquialisms and sentence structures have been improved, and syntax and grammatical edits I suggest here mostly fall into the "nitpicking" category, mostly in the newly written sections:

Line 153: The organization of these sentences is clearer than it was, but you still say the "four hirudinean orders" when it seems to me there are just the three, and two examples, Hirudo and Glossophina (sorry, review won't let me do italics) within the order Hirudinida.

Starting on Line 180, and then in other calls to the supplemental figures and the supplemental file itself... the first two "main" figures show Bayesian then Maximum Parsimony analysis. Then, for the two alternate scenarios the figures are presented Maximum Parsimony then Bayesian. If you wanted, you could reorder the alternate scenarios to match the main scenario. It's just something I noticed.

I noticed this on Line 256, and it pops up several other times, wherever you have new text, M. siluricus isn't italicized. It happens with a couple of other taxa too. Be sure to correct that.

On line 258, you noted that they do not bear limbs, right after you said they don't have lobopodous limbs. Did you mean arthropod-like limbs?

Lines 273 to 275, thank you for including this! I think it will help convince people its a sucker.

Line 293, it should read "annuli of Branchiobdellida"

I would move the "At the twist" sentence of lines 313 to 315 to before the "The fact that" sentence from lines 310 to 313, so that way the "The fact that" sentence summarizes the points from the previous three sentences, and has more of an "aha" impact.

Line 318: what do you mean by the "striations?" I can see that the dorsal side is a little more robustly preserved than the ventral side, but the only striation-like features I can see are the annuli.

Line 330 to 331: The clause here is a bit awkwardly worded, but still clarifying compared to the first draft.

Line 359: I would call to Fig. 3D as an example of the best maybe molt

Line 381: I would replace "cuticular body plan" with "annelid histology" to give it a more taphonomic flavor.

Line 389: I think you meant supplemental file 2.

Line 407: I think it should be Fig. 2A, not just 2.

Line 442: Are the Hirudiniformes Hirudinida + Acanthobdellida? I haven't seen that term before

In the Figure 1 caption, you still refer to the specimen number by its old designation, and not its updated UWGM number. I also noticed for the photographs that some of them have their museum abbreviations spelled out, or at least have references listed where you could find them, while others, like 'KUZ,' are more mysterious.

In the Supplemental Information .docx file, under character 67, the last "it" in the last sentence shouldn't be capitalized.

The literature references are improved, as is the background treatment. The references themselves still need to be checked for consistency of formatting. A few outstanding questions that I had:

On line 85: do you mean that cocoon building in general predates the Clitellata (such that it occurs in polychaetes and/or other phyla)?

On line 282... I'm still a little confused as to how the segmentation is externally visible. Is it because of the repetition of metameric circular organ/tubercle?

The article is also logically structured with good figures and shared raw data. The results are self-contained with relevance to the hypotheses. I noticed a little error in one of the subfigures, although I don't know if it can be fixed at this point: the tubercles in the inset of 1G do not match that of the broader diagram: the annulations in the inset show 1-4-0-1-4-2, not 1-4-2-1-4-2 as they should.

I still would've liked a low-angle photograph of the fossil to show how the putative tubercles stand out from the other taphonomic markings, but if its difficult to photograph or if you no longer have the specimen in your possession, that would be understandable. More on the tubercles below.

Experimental design

This is a good research article within the aims and scope of the journal as descibed in my first review, and considering that the discovery is one of potential considerable evolutionary significance, I would say it is relevant and meaningful.

I particularly wanted to thank the authors for including the extra phylogenetic analyses with the excluded poorly-known phyla, and attempting the phylum level analyses. I can understand not including it given the lack of resolution, but it is encouraging that the one time Macromyzon emerged from the polytomy was when a hirudinean was included. It also sounds like the authors are fully aware of the limitations and compromises with the data set. The only change I would consider making is to emphasize in the text that the phylogenetic analysis is there as a complement to the morphological designation of the taxa, rather than being the primary focus, mirroring what they said to me in the comments. I think our automatic go-to these days is to see what the phylogeny says, rather than the morphology (even though the morphology determines the phylogeny...).

Validity of the findings

The impact and novelty are not explicitly addressed, as is appropriate. All the underlying data have also been provided, and considerably expanded from the previous submission. The inclusion of the bayesian and maximum parsimony trees for alternate coding arrangements in an easy-to-access format is much appreciated, as is the coding information and rationale on the new characters from the phylogenetic matrix. That has gone a long way to strengthening my confidence in the phylogenetic anlayses of the paper. However, within the supplemental data file there are quite a few references listed as numbers, and others included as author lists. It looks like a references section was deleted at some point, and I think that will have to be added back in, with in-text references synchronized to be either author lists or numbers.

The conclusions are well stated and linked to the research question without going off on tangents. The expanded discussion on why the age point calibrations are as discordant with the fossil as they are is a good addition, and provided some context for me. The comparison sections with the other potential taxa is well-written and convincing, and I am thankful you did not include a comparison section to the Vetulicolans as reviewer 1 suggested. That would've raised more questions than it answered. I did have two suggestions for clarification:

For the last sentence of the paragraph on lines 468 to 471, you undermine your own taphonomic argument here in the "habitat" section since Macromyzon raises the possibility that hirudineans or clitellates evolved in marine or marginal marine waters. Maybe add an "although... Macromyzon may suggest that hirudineans aren't largely restricted to these habitats after all" hook somewhere in the sentence.

In the sentence from lines 537 to 539, you suggest that Macromyzon was most likely predatory (a new addition from the previous draft), but in the next paragraph you only talk about blood-feeding, parasitic, and commensal relationships with the arthropods. Maybe note the lack of specialized structures for vertebrate blood-feeding and suggest that they "fed" on arthropods, leaving the manner of trophic interaction a little more ambiguous (I could see them eating little Ceratiocaris or Acheronauta mid-molt).

Additional comments

Discussing the tubercles, as I couldn't really think of where in the above three categories this should fit... they terminologically go through a few transformations, although it is much better handled than in the first draft. I like the way they are presented in Figure 1, but in the Description, you introduce them as "semicircular protrusions" and then don't call them that again. Since this is the description, not the diagnosis, I would say here "...covered with semi-circular protrusions, which we interpret as metameric circular organs," just to link it to the Remarks.

Additionally, thank you for the clarity on the significance of the tubercles and why they weren't included in the phylogenetic analysis. I can see that as being a little too esoteric for inclusion in the text of the article. The fact that they aren't key for the phylogenetic placement makes me feel a little better about there not being a low-angle photograph that shows them if they're not as vital. However, their distribution does seem to be critical for the interpretation of the segmentation pattern of the organism, as I take it that is the only direct evidence of the sexannulate segmentation?

---

## Round 0.3 · Minor Revisions

Dear Authors, Please take into account the reviewer's request of making the statement a bit less "final". I think it would be a fair approach in a situation when two experts have contradictory opinions, especially regarding fossils. It will not diminish the value of your study, but provide a room for future interpretation. I hope you agree with me and we can accept the manuscript after these minor changes. Best regards, Dagmara

Reviewer 1 ·

Basic reporting

No comment

Experimental design

No comment

Validity of the findings

As the authors note in their response, the interpretation of this fossil material is subjective, and requires a number of interpretative steps that are not unambiguously supported by the data presented, or by clear precedent set by other material.

The authors have presented the evidence as best they can, and clearly feel that they have made a strong case for their preferred interpretation. I do not question the scientific value of a careful systematic description of the fossil material, and I have no objection to the authors’ setting forth their preferred interpretation. Readers will no doubt draw their own conclusion based on the evidence presented.

However, it would be invalid and irresponsible to present this material as an "unambiguous" leech body fossil. The authors’ interpretation is at best supported by ambiguous evidence, and in the worst case is a profound misinterpretation of the fossil material.

In order for a non-specialist reader to reach a balanced evaluation of the data and its implications, it is essential that those aspects of the study that are unambiguous and beyond reasonable objection (such as aspects of the fossil description) are clearly distinguished from aspects that represent the authors’ subjective interpretation of the fossil material.

Specifically, uncertainty in the interpretation of the fossil must be properly reflected in the manuscript. For example, a title such as “A putative leech body fossil predates…” would be supported by the material presented. Assertions of “clear evidence” might be more correctly phrased as “potential evidence”. Claims based on the authors’ interpretation should be prefixed with “If Macromyzon is a stem-group leech,”. Fig. 3 would not “refute” an ecdysozoan affinity, even if palaeoscolecids were the only group of vermiform ecdysozoans. The manuscript should articulate that phylogenetic results are contingent upon the authors’ chosen interpretations of the fossil material.

The systematics should only refer to aspects of the fossil that are securely demonstrated. The diagnosis should refer only to objectively observable morphological characteristics; it should certainly not refer to the genus as a Hirudinean (a subjective assessment); and it would be better to refer to the ‘caudal sucker’ in morphological rather than interpretative terms. Given that the interpretation of the “sucker” is not secure, it is also questionable whether it should be incorporated in the formal name of the genus.

The manuscript could be reframed to display a degree of caution commensurate with the uncertainty in the fossil information without great effort; and a more honest presentation of the strength of the data with an explicit acknowledgement of its potential shortcomings would if anything increase its potential impact, and ensure its continued relevance if future discoveries cause aspects of the interpretation to be revised.

Reviewer 2 ·

Basic reporting

This is my third time reviewing this manuscript, and it continues to improve with each iteration. Most of my recommendations and critiques at this point revolve around requests for clarification and suggestions for improvement, particularly of sections that have been added since the last revision. Most of these will be quick, and require no new analyses or figure redesigns on the part of the authors.

Line 132: This sentence feels like it is accidentally missing a verb. "we performed phylogenetic analyses USING Bayesian inference..."? Similarly, on line 137, you missed a period at the end of the sentence "...as further detailed in the 'Remarks' section below"

Line 163: I would capitalize "supplementary information" here, to indicate that it belongs to THE supplementary information file, rather than the supplementary information, inclusive of the tree and matrix files, more generally. Its unfortunate that we don't have better terminology available to distinguish between the two, here.

Also, I noticed that the supplementary figures and files have been reordered to follow a "bayesian -> parsimony" ordering, except for in the file names themselves in the supplemental data folder! Here, it still is listed "parsimony -> bayesian" for files 4 through 7.

Line 261: Perhaps just say "...associated with these segments that would suggest panarthropod affinity." The comma, then clause started by "which" makes it a little ambiguous to me as to whether the suggestion of panarthropod affinity is linked to the jointed or lobopodous limbs, or the fact that Macromyzon lacks evidence for them (even though I know its the former possibility).

Lines 296 and 297: you need a comma after "secondary," and the next sentence says "Typically" twice!

Line 305: I would start a new paragraph with the sentence "This leech-like annulation pattern..." its sufficiently different enough, and there is a decent amount of verbiage dedicated to it.

Line 309: I would mention "left and right margins," so as to eliminate any potential confusion with anterior/posterior margins.

Line 409: When you say the "same topologies" are recovered for the second set of supplementary analyses, do you mean the same as the original analyses or the first set of supplementary analyses, where both of the characters were coded as 'unknown?'

In Figure 1, at the start of subfigure E's description, I would say "Schematic of segmentation pattern of M. siluricus." for consistency with other illustration subfigures. In Figure 2, subfigure D, I take it that you mean "National Museum of Wales" in parentheses? Unless it is just called National Museum Wales?

Experimental design

The research still fits with the Aims and Scope of the journal, and the research question, namely why this Waukesha fossil is most likely a leech and the significance of that discovery, is still well-defined, relevant, and meaningful. I am looking forward for this information to get out into the broader paleontological community so we can look for more of these. The study is well-researched and investigated to a high technical and ethical standard, and most of my further comments pertain to the description of the methods and experimental design, in order to maximize reproducibility.

Line 144: To say that the inclusion of these taxa isn't necessary because Macromyzon doesn't belong to any of them isn't the best argument. Its always important in phylogenetic studies to have one or more plausible outgroups, but the reason why these taxa wouldn't even make good outgroups is because they are so controversial in their relationships and morphologically poorly constrained. I think that is the best reason not to include them: not that Macromyzon couldn't belong to them, but that they would only obfuscate, not clarify, any cladogram (your first point), and that for these reasons they don't make for great outgroup taxa.

In lines 145-158, you say that you have added a number of new taxa to the cladogram, and you list and describe them as well. For the sake of reproducibility, you should include at least the primary reference you consulted in order to determine their morphological character states in your character matrix.

From lines 146 to 154, you mention five polychaete genera and say that you performed an a priori analysis that returned the "correct" relationships between these taxa. I take it that this wasn't a separate study or a previous study, but that you did so with your character matrix that you used for Macromyzon, before Macromyzon was added? Or am I mistaken? Please clarify!

Line 179: After this sentence here, I would mention what the character states were that you used for your main analysis for characters 61 and 62, before discussing how you changed them for your supplementary analyses.

Validity of the findings

Comments on the validity of the findings are limited, and can be expressed with the following line calls. As before, the impact and novelty are not assessed, and all the necessary data have been provided either directly in the article or through supplemental material. The conclusions continue to be improved, and are generally within the scope of the provided evidence, carefully molded by what is laid out in the results and discussions.

Line 28: you've backed off the high levels of certainty you expressed in many of your statements compared to your first draft, and while I don't think it would be a good idea to go as far into ambiguity of interpretation as the first reviewer would like, saying that "The findings indicate that the earliest true leeches were marine" still feels a bit strong compared to the tone you've now adopted in the rest of the manuscript. 'Suggest' would be better than 'indicate,' I think. While this leech is marine, whether or not it truly is the earliest leech, and if all early leeches shared a marginal marine habitat, remains to be seen.

Line 87: I would say "little definitive insight" in regards to who made the cocoons. To say little insight here kind of dismisses the contribution they could make, but I think that pointing out the potential advanced-leech affinity of the cocoons later in the Discussion is one of the best pieces of evidence you have that the Triassic anchor point for estimating the appearance of leeches is a great underestimate of the true age of the first leech.

Lines 223 to 225: After this sentence here, I would suggest a call to Kluessendorf 1990, with respect to how the dissertation goes into more detail on the supratidal to subtidal nature of the Brandon Bridge Formation. The marginal marine nature of the lagerstatte will come in to play later with a suggestion I have for discussing the utility of the cocoon synapomorphy in the earliest leeches.

Line 303: In your response to me, you mention the taxonomic significance of the height of annuli, but this is not discussed in the manuscript itself, only the depths of sulci between segments and annuli. I would either explicitly mention the significance of annuli height earlier, or eliminate the reference to it here.

Lines 318-319, 323-324: This is the only part of the manuscript that still confuses me. You seem to suggest here that the caudal sucker is ventrally oriented, and that this is important for determining the animal's orientation, but earlier, on line 276, you mention that the sucker of Macromyzon is terminally oriented, and that terminal suckers can point anteriorly or posteriorly, but make no mention of dorsal vs. ventral. Was this a typo earlier, as to the orientation of Macromyzon's sucker?

Line 526: Here is where I think it is worth bringing up the treatment of Kluessendorf 1990 on the marginal marine nature of the Brandon Bridge Formation. Such an environment, I think, would actually still be compatible with the evolution of protective Clitellate cocoons, as supra- to subtidal environments can be quite harsh, prone to desiccation, temperature swings, and salinity swings. There isn't much sedimentological evidence for hard dessication at the Brandon Bridge, but it seems likely that short periods of exposure were likely in many places, and the environment could be very harsh, to the exclusion of most (unprotected) metazoans.

Line 541: Are these anticoagulants specific for vertebrate blood? Or do we not know that much? If all hirudinians, even acanthobdellidans and branchiobdellidans, have them, that would certainly suggest sanguivory somewhere in the deepest hirudinian ancestry. If that's the case, I hope that they are not vertebrate specific, as that could be problematic for your hypothesis, here.

Line 543: I notice here you mention "hirudinidan" excluding Acanthobdellida and Branchiobdellida. That could open up that sanguivory is a synapomorphy of crown group true leeches, acquired after Macromyzon. Just a thought.

Additional comments

Thank you for including the discussion of how you can tell the difference between the segments and annulations, featured on lines 286 to 289. That was very helpful. I just wish it came through on the pictures! I know how difficult these can be to photograph, though, with respect to subtle topographical features.

I also really like the phrasing of the last sentence in the paragraph from lines 411 to 412. It sums the results of your supplemental analysis up nicely.

I don't have any further comments for your figures or supplemental files, other than the few formatting and grammar corrections listed in the above sections. This is looking great!

---

## Round 0.4 · Minor Revisions

Thank you for taking into account all reviewers' comments.

The manuscript is close to being suitable for acceptance but the Section Editor noted:

> i feel the authors need to embrace more clearly the putative nature of their assignment in the title and abstract (e.g., the title and abstract sound more final then the actual rest of the manuscript; stem leech may be more appropriate and also seem to fall in the range of divergence estimate for molecular analyses). Only one of the reviewers (who is also the most critical one) has expertise on (fossil) annelids. Various important references dealing with evolution of annelids also seem to be missing (Parry et al. 2014; Bomfleur et al. 2015). At least some have argued for an origin of clitellata in the Late Paleozoic (Parry et al. 2014). Also arguing for an affinity of this fossil to Hirudinida rather than others like Branchiobdellida remains open for debate - also placing clitellata origin in the Silurian may be less problematic that placing Hirudinida). One argument supporting their assignment is that coccoons in Triassic are rather assignable to derived leeches is also open for discussion (compare Bomfleur et al. 2015 who assigned at least some of these coccoons based on findings of spermatozoa similar to those known in Branchiobdellida in Eocene ones). Parry, L., Tanner, A., & Vinther, J. (2014). The origin of annelids. Palaeontology, 57(6), 1091-1103. Bomfleur, B., Mörs, T., Ferraguti, M., Reguero, M. A., & McLoughlin, S. (2015). Fossilized spermatozoa preserved in a 50-Myr-old annelid cocoon from Antarctica. Biology Letters, 11(7), 20150431.

Please address the Section Editor's comments in you next revision.

---

## Round 0.5 · accepted · Accept

Thank you for providing the changes in the manuscript and answering the Section Editor's concerns. I am happy with the current version and consider it ready for publication.